

# The Path to CAM6: Coupled Simulations with CAM5.4 and CAM5.5

Peter A. Bogenschutz[1,2], Andrew Gettelman[2], Cecile Hannay[2], Vincent E. Larson[3], Richard B. Neale[2], Cheryl Craig[2], and Chih-Chieh Chen[2]

[1]Lawrence Livermore National Laboratory, Livermore, CA
[2]National Center for Atmospheric Research, Boulder, CO
[3]University of Wisconsin-Milwaukee, Milwaukee, WI

*Correspondence to:* Peter Bogenschutz (bogenschutz1@llnl.gov)

**Abstract.** This paper documents coupled simulations of two developmental versions of the Community Atmosphere Model (CAM) towards CAM6. The configuration called CAM5.4 introduces new microphysics, aerosol, and ice nucleation changes, among others to CAM. The CAM5.5 configuration represents a more radical departure as it uses an assumed PDF-based unified cloud parameterization to replace the turbulence, shallow convection, and warm cloud macrophysics in CAM. This assumed
PDF method has been widely used in the last decade in atmosphere-only climate simulations, but has never been documented in coupled mode. Here we compare the simulated coupled climates of CAM5.4 and CAM5.5 and compare them to the control coupled simulation produced by CAM5.3. We find that CAM5.5 has lower cloud forcing biases when compared to the control simulations. Improvements are also seen in the simulated amplitude of the El Niño 3.4 index, an improved representation of the diurnal cycle of precipitation, sub-tropical surface wind stresses, and double Intertropical Convergence zone biases.
Degradations are seen in Amazon precipitation as well as slightly colder sea surface temperatures and thinner Arctic Sea Ice. Simulation of the 20th century results in a credible simulation that ends slightly colder than the control coupled simulation. The authors find this is due to aerosol indirect effects that are slightly stronger in the new version of the model and propose a solution to ameliorate this. Overall, in these relatively untuned coupled simulations, CAM5.5 produces a credible climate that is appropriate for science applications and is ready for integration into the National Center for Atmospheric Research's
(NCAR's) next generation climate model.

## 1 Introduction

The Community Earth System Model (CESM, Hurrell et al. (2013)) is a state-of-the-art climate model consisting of atmosphere, land, ocean, and sea ice components which exchanges information and fluxes from each component via a coupler. Formerly known as the Community Climate System Model (CCSM), CESM is developed at the National Center for Atmospheric Research (NCAR) in a collaboration between researchers and students from universities, national laboratories, and
spheric Research (NCAR) in a collaboration between researchers and students from universities, national laboratories, and other institutions. The CESM and CCSM have been used to study the climate of the past, ranging from paleoclimate epochs to the recent past, and to make projections of possible future climate change.



The first version of the CCSM was released in 1996 (Boville and Gent, 1998). In the last twenty years there have been five official versions of NCAR's climate model, with the last two known as CCSM4 (Gent et al., 2011) and CESM1 (Hurrell et al., 2013), both of which produced simulations for the Coupled Model Intercomparison Projects fifth phase (CMIP5, Taylor et al. (2012)). Each successive CCSM/CESM release represents a climate model where most (if not all) of the component models

have upgraded versions from their predecessor.

With each successive upgrade to a climate model, the changes made in each component model contributes to the change in the climate simulation from the previous version. This paper will focus on changes to the climate simulation in the CESM model, but where only one component model is modified with upgraded physics. Here we will focus on the atmosphere component, known as the Community Atmosphere Model (CAM). CAM has evolved greatly over the past generations of CCSM and CESM

in terms of the physical parameterizations and dynamical cores employed. CCSM4 used CAM4 (Neale et al., 2013), which was known for its improved representation of ENSO and relatively improved representation of the Madden Julian Oscillation (MJO). CESM1 used CAM5 (Neale et al., 2010), which is notable for its improved representation of low clouds as well as the first version of CAM to include a microphysics and aerosol model sophisticated enough to be able to simulate cloud-aerosol interactions with reasonable physical fidelity.

The purpose of this paper is to document coupled climate simulations with snapshots of developmental versions of CAM, leading up to CAM6, that will ultimately be used in CESM2. More specifically, CAM6 will differ from CAM5 in terms of the use of prognostic precipitation in the microphysics, a four mode aerosol model, and updated ice nucleation schemes. In addition, CAM6 will also replace the boundary layer, shallow convective, and cloud macrophysical parameterizations. With such sweeping changes made to the treatment of physical parameterizations, the model development and coupled simulation

tests were broken into various subversions to allow for an incremental assessment of physics changes.

Perhaps the most radical departure from tradition in CAM6, compared to other CMIP5 GCMs and previous versions of CAM, is the treatment of cloud and turbulence physics. Traditionally, most atmospheric GCMs employ "separate" physics parameterizations that are responsible for simulating a particular process. CESM2 will contain a version of CAM that will employ the so-called "assumed PDF" method (Golaz et al., 2002; Larson et al., 2002). The assumed PDF method is a third

order turbulence closure that is centered around a multi-variate probability density function (PDF) that also serves as a cloud parameterization. Oftentimes, the functional form of the PDF is selected to be a double Gaussian, to accommodate the parameterization of stratiform cloud as well as convective clouds. Thus, the assumed PDF is often referred to as a "unified" parameterization, meaning that it has the capacity to parameterize various atmospheric processes and regimes (i.e. boundary layer process, warm cloud macrophysics, and shallow convective processes) with one parameterization call.

The last decade has seen the advent of the assumed PDF method used in numerical models, particularly cloud resolving models (CRMs) and GCMs. Cheng and Xu (2008), Larson et al. (2012), and Bogenschutz and Krueger (2013) demonstrated that idealized CRM simulations of boundary layer clouds with the assumed PDF method were shown to be much improved when compared to the CRMs with low-order closure turbulence parameterizations. Cheng and Xu (2011) were the first to implement the assumed PDF method into a global model, which was the super-parameterized version of the Community

Atmosphere Model (SP-CAM, Khairoutdinov et al. (2005)). They showed that in short simulations with prescribed SSTs that



the version of SP-CAM with an assumed PDF method implemented in the embedded CRM was able to greatly improve the simulation of marine stratocumulus, which was a persistent problem in the default version of SP-CAM.

However, it wasn't until the work of Bogenschutz et al. (2013) and Guo et al. (2014) when serious efforts to implement the assumed PDF method into conventionally parameterized GCMs, that are used in Climate Model Intercomparison Project (CMIP)
simulations, first began. The assumed PDF method implemented was the Cloud Layers Unified by Bi-normals (CLUBB, Golaz et al. (2002), Larson et al. (2002)) parameterization into NCAR's CAM (coupling described in Bogenschutz et al. (2013)) and Geophysical Fluid Dynamic Laboratory's (GFDL's) Atmosphere Model Version 3 (AM3, Donner et al. 2011, coupling described in Guo et al. (2014)). In both CAM and AM3, the implementation of CLUBB represents a radical departure from traditional physical parameterizations used in GCMs. The CLUBB parameterization replaces the planetary boundary layer (PBL),
shallow convection, and cloud macrophysical parameterization schemes in both models and represents a "unified" parameterization that is responsible for treating boundary layer clouds and shallow convection with one parameterization. This has many theoretical and scientific advantages compared to traditional physical packages made up of several different schemes which may or may not be compatible with one another. Indeed, both Bogenschutz et al. (2013) and Guo et al. (2014) demonstrate an improved performance for the simulation of intermediate types of regimes, such as the stratocumulus to cumulus transition,
which is represented by one turbulence parameterization in the CLUBB version but typically represented by three separate parameterizations with default GCM physics.

Following the work of Cheng and Xu (2011), Bogenschutz et al. (2013), and Guo et al. (2014), there have been additional efforts to implement the assumed PDF method into super-parameterized and conventional GCMs in similar manners (i.e. Cheng and Xu (2015), Wang et al. (2015)). In addition, some work has examined the performance of CLUBB serving as a deep
convection scheme, thereby serving as a completely unified parameterization of turbulence and clouds. Guo et al. (2015) tested such a model for the AM3 version of CLUBB and found that CLUBB serving as a deep convection scheme resulted in a reasonable mean state climate with improved tropical variability when compared the baseline AM3 model. However, they also found that the simulation of tropical water vapor and ice clouds in the midlatitudes was degraded. The work of Thayer-Calder et al. (2015) used CLUBB as a deep convection scheme in CAM but also tightly integrated the interface between clouds and
microphysics by drawing Monte Carlo samples of sub grid variability of temperature, water vapor, cloud liquid, and cloud ice, and feeding the sample points into the microphysics scheme. This technique is also commonly referred to as the "sub column" approach. Their results showed a general improvement in model skill compared to the baseline CAM5 model for most variables, but a degradation in the skill of precipitation.

There is no denying that the development activity, implementation, and evaluation of assumed PDF methods in the last
decade has been on an exciting upswing. In fact, the collaborative efforts between authors of the aforementioned works and this current work, have culminated in an assumed PDF-based scheme (CLUBB) being selected as default physics for CAM6 and hence CESM2. Thus far, one commonality of all published work involving implementation of assumed PDF methods in GCMs has focused on simulations using prescribed SSTs. While this is a convenient and necessary first step in parameterization implementation and testing in a global model, the final test of parameterization development is validation in a fully coupled
GCM. In addition, this is the only way a truly apples-to-apples comparison can be made with the baseline GCM, which





was likely tuned to produce scientifically credible coupled simulations. A coupled simulation with a new cloud or convective parameterization must not only simulate a good mean state climate, but also produce a stable pre-industrial coupled simulation, reasonable variability for the El Niño Southern Oscillation (ENSO), realistic sea ice, and a credible historical simulation of the 20th century.

This paper will document the coupled climate simulations for two developmental versions of CAM, on the path towards CAM6, compared to CESM1. The first developmental version will include all of the "non-CLUBB" physics changes to CAM (i.e the prognostic precipitation microphysics, 4 modal aerosol model, ice nucleation, etc.). The second developmental version will turn on the CLUBB parameterization in addition to the changes made in the first developmental version. It should be noted that the purpose of this paper is not to document the coupled performance of CAM6 or CESM2 model, but to document the

changes that occur in the coupled system when major changes are implemented into the CAM physics. This paper will be organized as follows: section 2 will give a description of the model versions used in this study, while section 3 will describe the model set up and configurations. Results will be presented in section 4 and will focus on the mean state climate, variability, and credibility of the 20th century simulation. Finally, section 5 will provide a summary of conclusions and general discussion.

## 2   MODEL DESCRIPTIONS

The standard CAM5 physics package (Neale et al., 2010), which is used in the control model for this paper, will be referred to as CAM5.3, which is the atmosphere component currently used in CESM version 1. These are the physical parameterizations that were used for the CESM CMIP5 submission in addition to the CESM large ensemble (LE, Kay et al. (2015)). CAM5 represents a nearly complete overhaul in physical parameterization options from CAM4, with the exception of the deep convection scheme (Zhang and McFarlane, 1995; Neale et al., 2008; Richter and Rasch, 2008). The boundary layer scheme in CAM5 is based on

downgradient diffusion of moist conserved variables (UWMT; Bretherton and Park (2009)), the shallow convection scheme follows that of Park and Bretherton (2009) (UWSC), while cloud macrophysics is computed according to Park et al. (2014). The Morrison and Gettelman (2008) two-moment stratiform microphysics scheme for both liquid and ice is used in CAM5, using the ice closures as described in Gettelman et al. (2010). Aerosols are predicted according to Liu et al. (2012) and linked to the microphysics through the parameterization of liquid and ice activation of cloud drops and crystals on aerosols (Gettelman

et al., 2010).

We will also examine the coupled climate simulations for the first developmental version of CAM towards CAM6, known as CAM5.4. The purpose of CAM5.4 is to include physical upgrades to the CAM5 family and assess their climate effects, before CLUBB was turned on for the system. In addition, at the time of CAM5.4 development it was unclear if CLUBB would be included into future versions of CAM as the default scheme, as CAM-CLUBB coupled simulations were still being evaluated.

Changes from CAM5.3 to CAM5.4 include an upgrade from a diagnostic precipitation scheme (Morrison and Gettelman, 2008) to a prognostic precipitation scheme (Gettelman and Morrison, 2015), a new ice nucleation scheme (Wang et al., 2014; Shi et al., 2015) to better represent mixed-phase and cirrus ice nucleation, an upgrade from the three Modal Aerosol Module (MAM3) to a four mode version (MAM4) that includes the treatment of black carbon (Liu et al., 2015), the use of an additional two





vertical layers near model top for consistent level treatment between CAM and the Whole Atmosphere Community Climate Model (WACCM), an improved treatment of the dust emissions size distributions and dust optical properties (Albani et al., 2015), a fix to the energy formulation in CAM (Williamson et al., 2015), a change to the vertical remapping from energy to temperature in the finite volume dynamical core, and consistent topography files for the finite volume and spectral element

dynamical cores.

The CAM5.5 version of the model uses all the upgrades developed for CAM5.4, however the shallow convection, planetary boundary layer, and warm cloud macrophysics schemes are replaced with the CLUBB parameterization (Bogenschutz et al., 2013). Because CLUBB is currently a warm cloud parameterization, ice cloud fraction and coupling is closed using the current relative humidity based scheme in CAM, as described by Gettelman et al. (2010) and Bogenschutz et al. (2013). Besides the

changes to the physical parameterizations, CAM5.5 represents an inherently different coupling with the microphysics. For example, in CAM5.3 and CAM5.4 there are three separate microphysics schemes; the double-moment scheme for stratiform clouds, while each of the Zhang and McFarlane (1995) (ZM) deep convection scheme and Park and Bretherton (2009) shallow convection scheme contains its own simplified single-moment treatment of microphysics.

In CAM5.5, since CLUBB is a unified parameterization, the double-moment microphysics is applied for both the strati-

form and shallow convection, although the simplified single-moment microphysics is retained for the ZM deep convection. Therefore, not only does CAM5.5 represent a more unified treatment of clouds and microphysics but also a more consistent treatment of cloud-aerosol interactions. In addition, CAM5.5 couples CLUBB and the microphysics together with the same time step, as opposed to the "sequentially split" method that is traditionally employed in CAM5.3, CAM5.4, and most other GCMs (as described in Gettelman et al. (2015)). In other words, each time CLUBB is called with its five minute time step,

the microphysics is called and this loop continues until the 30 minute CAM physics time step has expired. This is to ensure that that cloud water is not entirely depleted in a single time step, which is often the case with the long time steps commonly employed with coarse-grid GCMs. Table 1 describes the difference in physical parameterizations between CAM5.3, CAM5.4, and CAM5.5.

It should be noted that CAM5.4 and CAM5.5 were tuned slightly differently in order to achieve top of atmosphere (TOA)

radiation balances. For instance, CAM5.4 in its untuned state produced a TOA radiation imbalance of -1.4 W/m$^2$ for the first ten years of a pre-industrial control run. Therefore, a tuning decision had to be made, and to compensate for this imbalance a parameter that controls the auto conversion of ice to snow in the double moment microphysics ($D_{cs}$) was increased to produce more high clouds to help warm the climate system. However, when the CLUBB parameterization was run on top of an untuned version of CAM5.4 the TOA radiation imbalance was +1.5 W/m$^2$, therefore tuning decisions independent of those made in

CAM5.4 had to be taken.

CAM5.5 tuning involved both a decrease to the $D_{cs}$ parameter to reduce high level clouds as well as a modification to some CLUBB parameters to increase low cloud cover to cool the climate. Chiefly, the main CLUBB parameter that is tuned for radiation balance is the "$\gamma_{coef}$ parameter", which influences the width of the individual Gaussian components of $w$ relative to the width of the overall PDF of $w$ (Larson and Golaz, 2005). Decreasing the gamma parameter helps to decrease the




vertical turbulent transport of the higher-order moments, thus the suppressed upward mixing tends to increase low-cloud cover. Implications of these tuning parameter decisions will be discussed in section 4.

It is important to note that both CAM5.4 and CAM5.5 were only tuned at this point to achieve: 1) a TOA radiation imbalance of $< |0.1|$ W/m$^2$, and 2) a stable (non-drifting) pre-industrial climate. Neither configuration was purposefully tuned, in its

coupled state, to achieve optimal Arctic sea ice distributions, sea surface temperatures, or optimal overall model skill score improvements. Of the tuning parameters that are shared between CAM5.4 and CAM5.5, the two configurations only differ in their value of $D_{cs}$. CAM5.4 has a value of 250 $\mu$m while CAM5.5 has a value of 160 $\mu$m.

## 3   MODEL SIMULATIONS

In this section we will compare and assess the performance of CAM5.3, CAM5.4, and CAM5.5 run in coupled mode. This

includes a comparison of pre-industrial and 20th century historical runs. All simulations presented in this section were run using 1-degree horizontal resolution and the finite volume (FV) dynamical core. Table 2 lists the coupled simulations performed with each configuration. The CESM-CAM5.3 simulations represent those used in the CESM large ensemble (Kay et al., 2015) and includes a 2,100 year long pre-industrial control simulation. From the CESM-CAM5.3 1850 control simulation, the first member of the CESM LE was started at year 402. The remaining members of the CESM LE were started from the first member

at year 1920 by applying round-off temperature perturbations.

The CESM-CAM5.4 pre-industrial simulation was initialized from the CESM-CAM5.3 control run at year 402 and run for approximately 106 years. Since CAM5.4 was seen as a transitional model in the CAM development process, a 20th Century run was not performed with this configuration. Like the CESM-CAM5.4 simulation, the CESM-CAM5.5 pre-industrial control simulation was also initialized from the CESM-CAM5.3 control run at year 402 and was run for 200 years. At year 150, a single

20th Century member was started. We recognize the relative shortness in the simulation length of the CAM5.4 and CAM5.5 control runs, however, since both simulations achieve a stable equilibrium by the end of their runs we would not expect the simulated mean climate results to change much with a longer simulation.

## 4   RESULTS

Section 4.1 will focus on the simulated mean state climates produced by the three configurations of CAM in coupled mode.

For the sake of completeness and continuity, comparisons between CAM5.3, CAM5.4, and CAM5.5 will focus on the pre-industrial control runs. Sections 4.2, 4.3, and 4.4 will present results on the ocean meridional overturning circulation, the El Niño Southern Oscillation, and Madden Julian Oscillation, respectively. In section 4.5 results and implications of the historical run performed for CAM5.5 will also be presented.





## 4.1 MEAN STATE CLIMATE

Figure 1 displays the surface temperature evolution from the 1850 fully coupled pre-industrial control run for CESM-CAM5.3 (i.e. the control simulation used for the CESM LE), CESM-CAM5.4, and CESM-CAM5.5 for the first 200 years of integration. The CESM-CAM5.3 run reaches a reasonable equilibrium after about 90 years, whereas the CESM-CAM5.4 run stabilizes

after about 60 years. The CESM-CAM5.3 run takes longer to stabilize because it was initialized from present day Levitus observations. CESM-CAM5.5 has a longer spin-up period than CESM-CAM5.4 but stabilizes after about 100 years. All three simulations achieve a top-of-model radiation imbalance of $< |0.1|$ W/m$^2$, which is what we strive for in these pre-industrial control runs. Both the CAM5.4 and CAM5.5 simulations stabilize at a temperature slightly warmer than the CAM5.3 control runs.

First, we will demonstrate the improved cloud simulation with each successive model version, focusing on the cloud radiation biases. The simulated shortwave cloud forcing (SWCF) biases, computed relative to the Clouds and the Earth's Radiant Energy System Energy Balanced and Filled (CERES-EBAF, Loeb et al. (2009)) can be seen in Fig. 2. These figures represent 25-year climatological averages from a stable period in each simulation. The overall results for the analysis shown in this paper do not depend on the period selected for the averaging for any simulation, provided that period occurs after the model stabilizes. It is

important to note that we are comparing pre-industrial model simulations to present day observations. Thus, we expect there to be a bit of an offset between the two due to positive cloud feedbacks in a warmer world. With each successive model version there is a noticeable reduction in the root mean squared error (RMSE) score for SWCF, in addition to improvements in the pattern correlation coefficient. CESM-CAM5.3 contains large errors over the southern ocean, in the sub-tropical stratocumulus to cumulus transition areas, and over the tropical continents. These have all been longstanding biases in CESM and most

previous generations of the Community Climate System Model (CCSM).

With the introduction of CAM5.4, there is a significant reduction in the SWCF biases over the southern ocean. The bias in CAM5.3 exists primarily because low-level clouds contain insufficient amounts of supercooled liquid (Kay et al., 2016). This bias has been greatly ameliorated due to the new ice nucleation scheme (Wang et al., 2014; Shi et al., 2015) as well as the new prognostic microphysics scheme (Gettelman and Morrison, 2015). Previous work (Hwang and Frierson, 2013) suggests

that an improvement in the Southern Ocean SWCF biases could potentially lead to an improvement in the simulated double inter-tropical convergence zone (ITCZ) bias, which we will discuss further in this section.

Going to CAM5.5 physics we see a further reduction in the SWCF RMSE. These improvements appear to come from the tropical continents where there is a significant reduction in the amount of reflected shortwave radiation. As will be shown later it appears the reduction of these biases is due to a shift in timing of the most intense convection to later in the afternoon, when

the sun angle is lower. In addition, there are also improvements seen in the transition from stratocumulus to cumulus, whereas both CAM5.3 and CAM5.4 appear to transition a bit too abruptly, CAM5.5 tends to have a more gradual transition. This is generally in agreement with the prescribed SST results seen in Bogenschutz et al. (2013), however we note that there are some differences compared to that work.



The simulated biases for the longwave cloud forcing (LWCF), also computed relative to CERES-EBAF observations, are displayed in Fig. 3. Similar to the results for SWCF, each successive model simulation manages to achieve a better simulation for this quantity, although the improvements are more subtle for LWCF. The correlation coefficient is the same for each model configuration, suggesting improvements stem from reduction in magnitude of the errors. CAM5.3 contains longstanding biases

of an underestimate of LWCF in the midlatitudes and an overestimate in the tropics, which is partially due to biases related to the double ITCZ problem. CAM5.4 produces a global mean of LWCF that is most comparable to CERES-EBAF observations, however this is mostly due to compensating errors in the regional biases. While CAM5.4 improves the midlatitude bias in the storm tracks, there is a large positive bias over the tropical oceans. This bias is largely due to the tuning of the auto conversion from ice to snow parameter ($D_{cs}$) in order to achieve a TOA radiation balance. With CAM5.5 the global mean LWCF is more

comparable to observations than CAM5.3, but lower than CAM5.4. However, in this configuration the large positive biases seen in the tropics in CAM5.4 are somewhat ameliorated in CAM5.5, which is responsible for the slightly lower RMSE score.

Precipitation biases, computed relative to the Global Precipitation Climatology Project (GPCP, Adler and Coauthors (2003)), can be seen in Fig. 4. Unlike the cloud forcing biases, where the skill score improved with each successive model version, the skill for precipitation remains generally unchanged for all models. This is also true for the pattern correlation coefficient

between the three configurations. However, there are notable differences in regional biases between the three configurations. CESM-CAM5.3 has an obvious double ITCZ bias in the southern hemispheric tropical Pacific ocean and this bias is worsened in the CAM5.4 version. This is interesting because Hwang and Frierson (2013) identified a potential link between southern ocean cloud biases and double ITCZ biases in CMIP5 models; the idea being that a model with minimal southern ocean cloud biases would mitigate the double ITCZ due to global energy arguments. CESM-CAM5.4 shows a great improvement of the

southern ocean SWCF biases (Fig. 2), however, also shows a worsened double ITCZ bias. Kay et al. (2016) show that a version of CAM5.3 with reduced southern ocean cloud biases did not result in an improved double ITCZ bias because the northward cross-equatorial heat transports reductions occurred primarily in the ocean and not the atmosphere.

The CESM-CAM5.5 configuration does show a modest reduction in the double ITCZ bias, both in the Atlantic and Pacific oceans, compared to the CAM5.4 and CAM5.3 versions of the model. It should be noted that all three versions of CAM use

the same deep convection scheme, therefore the slightly improved double ITCZ in CAM5.5 can be attributed to interactions of CLUBB with the ZM scheme and/or feedbacks to/from the coupled system with CAM5.5. We note that all tuning simulations we performed with CAM5.5 (not shown) resulted in a reduced double ITCZ bias of varying degrees when compared to CAM5.3 and CAM5.4. Thus, it does not appear that this was achieved simply due to happenstance. Other regions of improved precipitation in CAM5.5 can be found over the subtropics, the Atlantic deep convective regions, and Australia. There are also

regional degradations in CAM5.5, such as over the tropical Pacific warm pool and the Indian Ocean.

The most notable bias is over tropical South America where precipitation is greatly reduced compared to observations and CAM5.3 and CAM5.4. It is interesting to note that the coupled simulation precipitation results for CAM5.5 are not dissimilar from those presented in the prescribed SST study of Bogenschutz et al. (2013), with the exception being over the Amazon, hinting at a possible feedback between moisture transport and the Pacific and Atlantic SSTs with this region (Martins et al.,

2015). However, An examination of the large-scale circulation over the area (not shown) provided no significant differences





between the CAM5.3 and CAM5.5 simulations, suggesting that the difference in precipitation simulation is likely linked to differences in parameterized physics as opposed to biases in the large-scale circulation.

For a more in-depth look at the precipitation biases over the Amazon for CAM5.5, we examine the diurnal cycle of precipitation for CAM5.3 and CAM5.5 from the pre-industrial control runs. We note that although CAM5.4 is not included in this

analysis, because sufficient output from the pre-industrial control run was not supplied, examination of the diurnal cycle of precipitation for CAM5.4 in shorter prescribed SST simulations has been performed and the behavior was shown to be nearly identical to that of CAM5.3. It is known that GCMs struggle to simulate the timing and intensity of precipitation (Dirmeyer et al., 2011). Fig 5 shows this is true for CESM-CAM5.3 over the tropical continents for DJF and JJA, with the peak precipitation occurring around noon, whereas the Tropical Rainfall Measuring Mission (TRMM, Huffman et al. 2007) observations

generally show a peak around 19:00 local time. Thus, the CAM5.3 simulation of precipitation is tied too closely to the peak of solar insolation. The CAM5.5 representation, on the other hand, shows a large improvement when compared to CAM5.3, with the peak precipitation generally occurring around 17:00 local time. The reason for this improvement appears to be coming from the CLUBB parameterization, which is responsible for the growth of the boundary layer and mid-morning and early afternoon shallow convection. The CLUBB unified parameterization is able to successfully simulate a gradual transition of these regimes

and prevent the deep convective scheme from firing off too early.

Figure 6 (bottom) shows the composite of the precipitation over the Amazon for the DJF season for CAM5.3 and CAM5.5. Both CAM5.3 and CAM5.5 underestimate the peak precipitation rate when compared to the observations, however CAM5.3 begins to precipitate much too early in the day compared to the observed time. Therefore, CAM5.3 has a better mean state bias in the Amazon, but for the wrong reasons, since it begins to precipitate too early but ends at approximately the same time

as CAM5.5. Improvements to CAM5.5 mean precipitation should therefore focus on increasing the intensity and duration of the precipitation, since both CAM5.3 and CAM5.5 stop precipitating too early. Experiments and modifications to the deep convection scheme are currently underway. It is, however, encouraging that CAM5.5 is able to improve the diurnal cycle of precipitation over tropical land as this has been a longstanding bias in GCMs. Notably, this has been achieved without changing the deep convection scheme. Further improvements in the representation of the diurnal cycle of precipitation may be achieved

by removing the conventional deep convection scheme and allowing CLUBB to also simulate this regime (Thayer-Calder et al., 2015).

The sea surface temperature (SST) biases, computed relative to the pre-industrial HadISST observation estimates (Rayner et al., 2003), for the three models are shown in Fig. 7. Unlike other variables displayed in this section, which showed either an improved or static skill score compared to the previous model iteration, the SST shows worsening skill with each successive

model iteration. In a sense, this is not very surprising as the CAM5.3 configuration, which was used for the CESM large ensemble project, was well tuned to achieve very good SSTs. As mentioned in section 2, neither CAM5.4 or CAM5.5 was purposefully tuned to achieve better skill scores for any variable, but only tuned to achieve radiation balance and a stable pre-industrial coupled run. However, neither CAM5.4 or CAM5.5 are substantially worse than the CAM5.3 control and it is likely that either could be tuned/adjusted to match the CAM5.3 skill by slight adjustments to the cloud forcing via the

CLUBB/macrophysics tunable parameters and/or by exploring sensitivities to the ocean parameters.



With the introduction of CAM5.4 physics, a cold bias becomes present in the north Pacific and especially the north Atlantic Oceans. Likely this is due to the improved shortwave cloud forcing and more reflective clouds. This is also the case in the Southern Ocean. Why sea surface temperature biases get worse with improved regional cloud forcing is unclear and warrants further investigation to a potential correction of a compensating bias in the ocean model. While biases in the midlatitude regions

between CAM5.4 and CAM5.5 are very similar, differences do exist in the tropics and subtropics. For example, CAM5.4 has a large positive SST bias in the southeastern tropical Pacific, which is ameliorated in CAM5.5 and one of the likely reasons for the reduced double ITCZ bias. Similar bias reductions are found in the tropical Atlantic Ocean. Not surprisingly, tropical SST biases in CAM5.5 are well correlated with the precipitation biases, namely over the tropical western Pacific and western Indian Oceans.

Surface stress from the atmosphere is another important component to the coupled system and the surface stress biases, computed relative to the European Remote Sensing Satellite Scatterometer (ERS, Bentamy et al. (1999)) observations, can be seen in Fig 8. Similar to the SST biases, we see that the positive surface stress bias in the southern Ocean gets worse in CAM5.4 and CAM5.5 when compared to CAM5.3, which is a potential reason for the degradation seen in SST. This degradation is likely due to vast changes in low clouds over these regions and reconciling these changes for an improved representation of surface

stresses and SSTs is an area left for future work. The addition of CAM5.5 physics, CLUBB, neither improves nor degrades these biases, suggesting they are the result of the addition of the CAM5.4 physics. An examination of the surface stresses in the subtropics, where boundary layer clouds and trade wind cumulus are prevalent, shows that CAM5.5 ameliorates much of the positive bias seen in the surface stress magnitude in CAM5.3 and CAM5.4.

Figure 9 displays the biases for Arctic sea ice computed relative to the HadISST pre-industrial dataset. CESM-CAM5.3

generally has the best agreement with observations and this is not surprising since the sea ice model was not tuned at all in the CESM-CAM5.4 and CESM-CAM5.5 simulations. While the two new configurations of CAM tend to produce less sea ice in the Arctic and more in the North Atlantic and Labrador sea compared to the baseline CESM-CAM5.3 configuration, it is likely that either could match the quality of the baseline simulation with improvements made to either the sea surface temperatures or tuning to the sea ice model. However, it does appear that CESM-CAM5.4 produces the thinnest sea ice in the Arctic, while

CESM-CAM5.5 is only marginally lower than the baseline CESM-CAM5.3 simulation. All three configurations use the same generation of the Community Ice CodE (CICE), thus differences can be attributed to differences in atmospheric physics or tuning of the atmosphere model.

## 4.2 OCEAN MERIDIONAL OVERTURNING CIRCULATION

Figure 10 shows the global and Atlantic Ocean meridional overturning circulation (MOC) for CESM-CAM5.3 and CESM-

CAM5.5. The maximum overturning in the Atlantic occurs near 35° N at a depth of 1 km for both CESM-CAM5.3 and CESM-CAM5.5. CESM-CAM5.5 is weaker at about 23 Sverdrups (Sv = $10^6$ m$^3$s$^{-1}$), compared to CESM-CAM5.3 at about 26 Sv. For comparison, CCSM3's maximum Atlantic MOC (AMOC) was about 20 Sv (Collins et al., 2006), whereas the maximum AMOC in CCSM4 was 24 Sv (Gent et al., 2011). However, whereas these configurations used different mixing parameterizations in the ocean model, the ocean models in CESM-CAM5.3 and CESM-CAM5.5 are largely the same. There-



fore any differences can be linked to differences in the atmosphere models. The likely reason for the differences between the two configurations is in the simulation of the surface wind stress (Fig. 8) in the north Atlantic. Whereas CESM-CAM5.3 and CESM-CAM5.4 contain positive biases in the Labrador Sea, this has been largely reduced in the CESM-CAM5.5 simulations. Overall, however, it does not appear that the inclusion of CLUBB degrades the simulation of AMOC in CESM-CAM5.5.

Observational estimates of MOC can be found in (Orsi et al., 1999).

## 4.3 EL NIÑO SOUTHERN OSCILLATION

Of great importance for a climate model to represent with some fidelity is the El Niño-Southern Oscillation (ENSO), which is the strongest coupled mode of variability in the climate system. Teleconnections from the warming of the tropical waters in the eastern Pacific, associated with El Nino events, have significant impacts on weather and climate over much of the

planet, which illustrates the importance to represent in a climate model. Previous versions of NCAR's climate model, namely CCSM3 (Collins et al., 2006), struggled to simulate ENSO, which was dominated by variability at 2 yr, rather than the 3-7 yr period from observations. The ENSO period was greatly improved upon with CCSM4 (Gent et al., 2011), with the introduction of changes made to the deep convection scheme (Richter and Rasch, 2008; Neale et al., 2008), however the simulated ENSO in CCSM4 still had an unrealistically large amplitude.

Figure 11 displays the variance spectra of the Niño-3.4 monthly SST anomalies for observations (HadISST) and for CESM-CAM5.3, CESM-CAM5.4, and CESM-CAM5.5. Various 100-year samples are displayed from the long CESM-CAM5.3 control and it is quite clear there exists variability in the amplitude and periodicity of the simulated ENSO within this long control run. Therefore, caution must be exercised when evaluating the relatively short control simulations (Wittenberg, 2009) of CESM-CAM5.4 and CESM-CAM5.5. However, it is also clear that the amplitude of the simulated ENSO from CESM-CAM5.4 is

unrealistically large. Obviously, this simulation sparked concern about an inherent deficiency in the CAM5.4 physics causing a degradation in the simulation of ENSO. Sensitivity experiments revealed that the tuning of $D_{cs}$ for radiation balance for CAM5.4 was the cause of the large amplitude ENSO.

The ENSO simulation provided by CESM-CAM5.5 is much more reasonable than CAM5.4. While CESM-CAM5.5 simulates a better amplitude compared to CESM-CAM5.3, the periodicity is on the shorter end, but still acceptable. It should be

noted that the first 100 years simulated by CESM-CAM5.3 also had a three year periodicity, but eventually settled into a four to five year periodicity, which is closer to observations. At this point, it is unclear if a longer simulation of CESM-CAM5.5 will result in slightly longer periodicity. However, it is worthwhile to note that it does not appear that the simulation of ENSO is significantly improved or degraded with the addition of the CAM5.5 physical parameterizations. The Niño-3.4 time series can be seen in Fig. 12 and demonstrates the ability of CESM-CAM5.5 to simulate, with reasonable fidelity, the variable cycles

associated with ENSO, including the often observed two year La Niña events that follow a one year El Niño event.

## 4.4 MADDEN JULIAN OSCILLATION

Although the deep convection scheme has not changed in the evolution of CESM experiments shown in this paper, it is still important to assess the differences in the simulation of inter-annual seasonal tropical variability in the simulations with the



new cloud and turbulence physics. Figure 13 shows the diagrams for the outgoing longwave radiation (OLR) and 850 hPa wind vector anomalies for the eight lifecycle phases associated with the Madden Julian Oscillation (MJO) for ERA, CESM-CAM5.3, and CESM-CAM5.5. The time periods displayed for the model simulations are the same as shown for the results on climatology in section 4.1.

The ERA analysis clearly shows eastward propagation of the OLR anomalies, associated with the MJO. CESM-CAM5.3 shows very little variability, characteristic of a model that struggles to simulate the MJO. While there is a slight improvement in the strength of the anomalies and signal of the propagation in the CESM-CAM5.5 simulation, it is still much weaker than in observations. While the lack of a significant improvement compared to observations is not surprising in CESM-CAM5.5, due to the MJO mainly being driven by the parameterized deep convection, further improvements of the MJO may be achieved by
allowing the CLUBB parameterization to also simulate the deep convective regime, such as the findings presented in Thayer-Calder et al. (2015), or by modifications to the existing deep convection scheme in CAM5.5.

## 4.5   HISTORICAL SIMULATIONS

Figure 14 displays the time series of the globally averaged surface temperature anomaly for 1920 to 2005 from observations and the ensemble mean from 30 members of the CESM-CAM5.3 configuration. The model spread from the CESM-CAM5.3
ensemble is denoted by the shading. One realization from the CESM-CAM5.5 model is also displayed. Once again, it should be noted that a decision was made not to run the CESM-CAM5.4 model for the historical simulation since it was seen as an intermediate model version along the CAM development process. In addition, we stress that only a single realization from CESM-CAM5.5 is shown. Thus, the point of examining this is only to gauge the interplay between the climate sensitivity and aerosol interactions in CESM-CAM5.5 and how they may come into play for the 20th century simulation. The CESM-CAM5.5
run was started from year 150 of the pre-industrial control. Details on the specifics of the initialization of the CESM-CAM5.3 model can be found in Kay et al. (2015).

    The single realization of CESM-CAM5.5 stays mostly within the model spread of the 30 member CESM large ensemble, with a couple exceptions. The first is a period in 1930 that is much warmer than observations and the CESM-CAM5.3 model average. The second is that the CESM-CAM5.5 simulation ends cooler than the CESM-CAM5.3 average and about as cold as
the coldest member. Although it is difficult to attribute these differences to either changes in the CAM physics or to internal variability, the fact that the CESM-CAM5.5 simulation ends colder than observations is worth investigation. We identify the three most likely reasons for this difference: 1) noise from internal variability that cannot be quantified from one ensemble member, 2) a relatively short pre-industrial control run in which the ocean may not be fully adjusted to the CAM5.5 physics, and 3) changes in the climate sensitivity and/or aerosol indirect forcing. Here we focus on reason three, since we can readily
quantify these measures.

    Various slab ocean model (SOM) experiments with doubled CO2 concentrations performed throughout the CAM-CLUBB development process have identified a climate sensitivity of 3.8 K associated with CAM5.5, which is slightly lower than the climate sensitivity of 4.1 K associated with CAM5.3 (Gettelman et al., 2012). These estimates are higher than the CMIP5 model mean $2xCO_2$ equilibrium climate sensitivity of 3.37 K (Andrews et al., 2012). While the climate sensitivity associated




with CAM5.5 is indeed slightly lower than CAM5.3, it is also necessary to investigate the compensating effect of aerosols on the climate system.

CAM5.3 has an aerosol indirect effect (AIE) that is too large when compared to estimates obtained through satellite measurements (Quaas et al., 2009). Table 3 documents the radiative flux perturbation (RFP), changes in SWCF and LWCF, and the AIE for various aerosol perturbation experiments involving several versions of CAM throughout the development process. All experiments shown in Table 3 represent an aerosol perturbation calculation where each configuration used climatological SSTs and present day (PD) forcing and was run twice; once with PD aerosol emissions and the other with pre-industrial (PI) aerosol emissions. Thus the values shown in Table 3 are the differences between the simulations using PD aerosol emissions minus PI emissions. The RFP (Lohmann et al., 2010) is defined as the difference in top of model (TOM) radiation imbalance between the PD and PI aerosol emission simulations. The AIE is defined as AIE = $\Delta$SWCF + $\Delta$LWCF.

CAM5.3 has an RFP and AIE that is larger than satellite estimates presented in Quaas et al. (2009). Therefore we can conclude that the successful CESM-CAM5.3 simulation of the 20th century is due to competing effects of a potentially large climate sensitivity and an AIE that is too strong. Gettelman et al. (2015) showed that the RFP and AIE is reduced by the implementation of the MG2 prognositc precipitation scheme (denoted by the CAM5.3+MG2 simulation in table 3). The reason being that precipitation processes are altered with more accretion relative to autoconversion in MG2. Accretion does not depend on cloud drop number, so the clouds are less sensitive to drop number. However, simulations of CAM5.3+MG2+CLUBB displayed an increase in the RFP and AIE when compared to the CAM5.3+MG2 simulations. This increase is due to the fact that since CLUBB is a unified parameterization of stratiform and shallow convective clouds and drives a single microphysics scheme, the aerosol indirect effect is now being considered in more cloud types than CAM5.3. While this physical consistency is desirable for a global model, it does subject the model to an increase in the sensitivity of cloud-aerosol interactions.

However, performing AIE experiments with CAM5.4 (which includes the MG2 prognostic precipitation scheme), we see that the forcing to aerosols has rebounded to CAM5.3 values. As expected, due to the AIE being considered in more cloud types than CAM5.4, the sensitivity is even higher for CAM5.5. Puzzled why CAM5.4 has an aerosol sensitivity so much higher than CAM5.3+MG2, the authors investigated and found that the increased sensitivity was due to increased lifetime of SO2, which was due to the new MAM4 aerosol model. Nevertheless, we now gain a deeper understanding for why CESM-CAM5.5 ended the 20th century simulation colder than observations and CESM-CAM5.3; due to compensating effects of a lower climate sensitivity and higher aerosol sensitivity when compared to CESM-CAM5.3.

The higher aerosol sensitivity associated with CESM-CAM5.5 is a combined effect due to the increased lifetime of SO2 from CAM5.4 physics as well as the fact that the AIE now being considered in more cloud types with CAM5.5. One potential solution, following Gettelman (2015) is to change the autoconversion and accretion process rates from the default used in the MG2 microphysics (Khairoutdinov and Kogan, 2000) to that of Seifert and Beheng (2001). Seifert and Beheng (2001) has lower autoconversion rates for lower liquid water paths than Khairoutdinov and Kogan (2000) because it includes a hysteresis effect whereby autoconversion in the absence of existing rain is delayed, thus damping the AIE in the shallow cloud regime that CLUBB and MG2 are now acting on in CAM5.5. Indeed, Table 3 shows that in climatologically prescribed SST simulations, the CAM5.5 runs using the Seifert and Beheng (2001) autoconversion and accretion physics does reduce the AIE and RFP. The





authors are currently investigating coupled simulations of CESM-CAM5.5 with the new process rate calculations that will be included in a future version of CAM6.

## 5    Conclusions

In this paper we documented coupled simulations from various configurations of CAM along the development track towards
CAM6. The baseline simulation is CESM-CAM5.3, which uses the model physics used for the CESM large ensemble project and CMIP5 simulations. The CESM-CAM5.4 simulation updates many of the aerosol physics and microphysics parameterizations in the model, while CESM-CAM5.5 updates the turbulence, shallow convection, and boundary layer physics. More specifically, CAM5.5 represents the implementation of the CLUBB parameterization. While the CAM5.5 model represents the physics package likely to be used for CAM6, it should be noted that CESM-CAM5.5 simulations documented in this paper do
not represent CESM2. The purpose of this paper is to document changes to the coupled simulations when only the atmosphere component is changed. In addition, this is the first time coupled simulations have been documented in a climate model using the "assumed" PDF method, which has been a method experimentally implemented into many atmosphere only climate models during the past decade. CESM2 will introduce new generations for the ocean, land, and sea ice models. It is also likely that CAM6 will differ slightly from CAM5.5 as the CAM model will need to be tuned with the newer component models. In addi-
tion, adjustments will have to be made to CAM5.5 to improve some of the degradations to the simulated climate introduced by CESM-CAM5.5 and the new component models.

Results presented in this paper focused on the pre-industrial control runs between the three CESM configurations. All three simulations were tuned to achieve a top of model radiation imbalance $< |0.1|$ W/m$^2$ and all three were able to achieve a stable pre-industrial control. The largest improvement seen with each successive CAM configuration was in the simulation of
clouds, as seen through the cloud forcing biases. This is not so surprising since the major upgrades to the CAM model between CAM5.3 and CAM5.5 are in the cloud parameterizations. Big improvements are seen in the Southern Ocean, tropical land, and the stratocumulus to cumulus transition regions. These improvements are due to combined effects of the CAM5.4 physics upgrades and the CLUBB parameterization (CAM5.5).

CAM5.4 and CAM5.5 were tuned only to achieve a top of model radiation balance and a stable pre-industrial control
run. Therefore, no effort was put forth to attempt to achieve the best possible SSTs or sea ice distribution, which is often done when a model is prepared for a major release, as CAM5.3 has been. With that said, it is therefore not surprising that the simulation of SST is somewhat degraded for CESM-CAM5.4 and CESM-CAM5.5 when compared to CESM-CAM5.3. Most of these degradations are coming from the midlatitude storm tracks, where there has been a substantial improvement in the representation of clouds. Therefore, it will be interesting to assess the potential of compensating errors in CESM-CAM5.3.
Nevertheless, the fact that both CESM-CAM5.4 and CESM-CAM5.5 only slightly underperform compared to CESM-CAM5.3 for both SST and sea ice is rather remarkable, given the drastic change to the physics in these versions of CAM. A bit of tuning and integration with the new component models in CESM2 should reconcile the slight degradation seen in the simulation of these variables.



The overall simulation of the mean climatology of precipitation was decidedly mixed with the new version of CESM-CAM5.4 and CESM-CAM5.5, as both contain regional improvement and regional degradations. For instance, CESM-CAM5.4 tends to exacerbate the double ITCZ bias. Since the Southern Ocean clouds are also improved, this indicates that in coupled model simulations South Ocean cloud biases may not influence the ITCZ biases. On the other hand, CESM-CAM5.5 does

tend to have a slightly improved double ITCZ bias, when compared to CESM-CAM5.3, but it also contains a dry bias over the Amazon rainforest. We have determined that this precipitation bias in the Amazon was due to improving a compensating error in CAM5.3. That is, CESM-CAM5.5 tends to have better timing of the most intense precipitation over tropical land than does CESM-CAM5.3. CESM-CAM5.3 tends to precipitate too early over land, a common GCM bias, whereas CESM-CAM5.5 starts to rain at roughly the correct time, it stops too early. We conclude that efforts to ameliorate this dry bias in CESM-

CAM5.5 should focus on generating more intense and longer duration precipitation events over the Amazon and modifications to the deep convection are currently underway.

Minor improvements can also be seen in the simulation of the MJO, as CESM-CAM5.5 tends to have a bit more low frequency variability in the eastward propagating convection. This is despite using the same convection scheme in all three versions of CAM. However, the simulated MJO in CESM-CAM5.5 is still much weaker than the observed MJO. More substantial

improvements may be achieved in a future version of CAM, where we allow CLUBB to simulate the deep convection (Thayer-Calder et al., 2015), or by modifying the deep convective scheme.

Perhaps one of the most important simulated features in coupled simulations is the El Niño Southern Oscillation (ENSO). CESM-CAM5.3 has a reasonable ENSO simulation, with a period that agrees well with observations, but an amplitude that is considered to be too large. CESM-CAM5.4, on the other hand, exacerbates the amplitude bias. However, it was found through

sensitivity studies that the ENSO amplitude was directly related to how this model configuration was tuned for radiation balance. While CESM-CAM5.5 appears to improve the amplitude of ENSO compared to CESM-CAM5.3 and CESM-CAM5.4, it is hard to give a definitive answer due to the relatively short pre-industrial control simulation. CESM-CAM5.3 exhibits noticeable variability in the simulation of ENSO throughout its 2100 year control run, similar to Wittenberg (2009), thus caution must be exercised when analyzing ENSO from a 200 year simulation.

Another very important metric when assessing coupled model performance, is the credibility of the 20th century simulation. Any climate model with upgraded physics should be able to faithfully simulate the observed temperature trend of the 20th century, to give confidence of a credible simulation in the presence of aerosol and greenhouse gas forcing. Due to computing restraints, only one realization of the 20th century was performed for CESM-CAM5.5, thus we used this simulation as a way point in assessing the interplay between climate sensitivity and aerosol effects. For the most part, the simulated temperature

trend stays within the bounds of the thirty member CESM large ensemble. The simulated temperature anomalies for 2000-2005 do end up a bit on the cold side. Knowing that this may be cause for concern for some scientists in the CESM2 development process, we conclude that the most likely reason for this is a reduction in climate sensitivity in CESM-CAM5.5 compared to CESM-CAM5.3 (which has been an outlier in terms of CMIP5 models for this metric) and an increased cloud-aerosol sensitivity.





The reason for the increased cloud-aerosol sensitivity is two-fold; the first reason relating to an increase in SO2 lifetime with the introduction of CAM5.4 physics, due to the new aerosol model, and the second reason being that cloud-aerosol interactions are computed in more cloud regimes in CAM5.5 than they are in CAM5.3 or CAM5.4. We propose a solution to decreasing the aerosol-cloud sensitivity in CAM5.4 by switching the auto conversion and accretion physics to the formulation proposed

by Seifert and Beheng (2001), which tends to decrease precipitation autoconversion at low liquid water paths.

While this paper does not document the coupled simulations that will be produced by NCAR's next generation climate model (CESM2), it is important to document the coupled model performance throughout the development process to highlight where notable improvements and degradations are coming from. In addition, this paper represents the first time that coupled simulations have been documented from a model using the "assumed PDF" method for climate simulations. This is a method

that has been widely employed for experimental implementation into atmosphere only climate models, but is important to assess the feasibility of running such a parameterization in the coupled model.

While the simulation of the coupled climate is encouraging with CESM-CAM5.5, exciting development opportunities still lie ahead of us. By removing the deep convection scheme from CAM5.5 and allowing CLUBB to operate on this regime (i.e. Guo et al. (2015) and Thayer-Calder et al. (2015)) we would have a unified parameterization that could handle all cloud

and turbulence. In addition, this unified parameterization would drive a single microphysics scheme, allowing for a consistent treatment of cloud-aerosol interactions in all cloud types. Removing the deep convection scheme would also remove any undesired interactions between the ZM scheme and CLUBB to allow for a true assessment of the scale sensitivity of CLUBB for GCM simulations. While previous studies have already shown that these PDF schemes can function in a scale insensitive manner for CRMs (Bogenschutz et al., 2013; Larson et al., 2012; Cheng and Xu, 2008), some preliminary GCM studies (Guo

et al., 2014; Bogenschutz et al., 2013; Cheng and Xu, 2011) do show at least some sensitivity to horizontal and/or vertical grid sizes. However, it is unclear if these sensitivities stem from the traditional deep convection schemes in these models. Having one unified parameterization would add clarity towards assessing the scale sensitivity of these assumed PDF methods in GCMs simulations.

## 6 Code and Data Availability

The model code used in these simulations is stored within the CAM development repository and is available upon request from the corresponding author. Results in this paper are based on CESM tag cesm1_4_01_n27_cam5_3_77, which is not a publicly released version of CAM but is available on the CESM developers repository at https://svn-ccsm-models.cgd.ucar.edu/cam1/branch_tags/ca Access and terms of use to the CESM developer repository can be found at https://www2.cgd.ucar.edu/sections/cseb/development-code. Climatology files of model runs used to generate figures in this paper have been published at zenodo.com (doi:10.5281/zenodo.815593

*Competing interests.* The authors declare that they have no conflict of interest





*Acknowledgements.* Peter A. Bogenschutz is supported by National Science Foundation grant number 0968657. The National Center for Atmospheric Research is sponsored by the United States National Science Foundation. V. Larson gratefully acknowledges financial support under Grant 0968640 from the National Science Foundation and Grant DE-SC0006927 from the SciDAC program of the U. S. Department of Energy. The authors thank Katherine Thayer-Calder and Julio Bacmeister for comments and suggestions. This work was performed under the auspices of the U.S. Department of Energy by Lawrence Livermore National Laboratory.





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





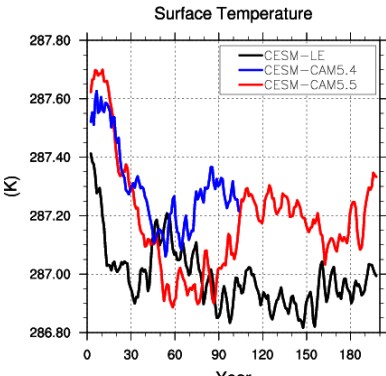

**Figure 1.** Evolution of the globally averaged surface temperature for the first 200 years of the pre-industrial control run for CESM-CAM5.3 (black curve, CESM Large Ensemble), CESM-CAM5.4 (blue curve), and CESM-CAM5.5 (red curve). CESM-CAM5.4 was only run for 106 years.





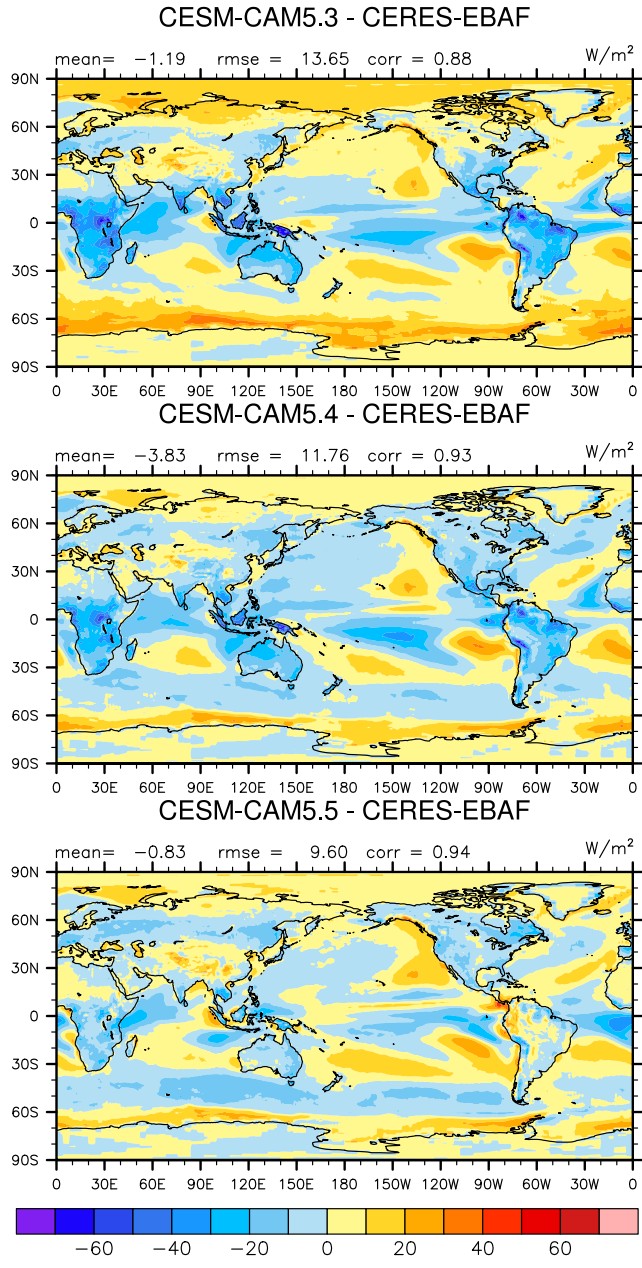

**Figure 2.** Shortwave cloud forcing biases as computed relative to CERES-EBAF for (top) CESM-CAM5.3 (years 402 to 426), (middle) CESM-CAM5.4 (years 75 to 100), and (bottom) CESM-CAM5.5 (years 100 to 125) for the pre-industrial control run. Each configuration displays the difference from the observed mean, root mean squared error, and pattern correlation coefficient.



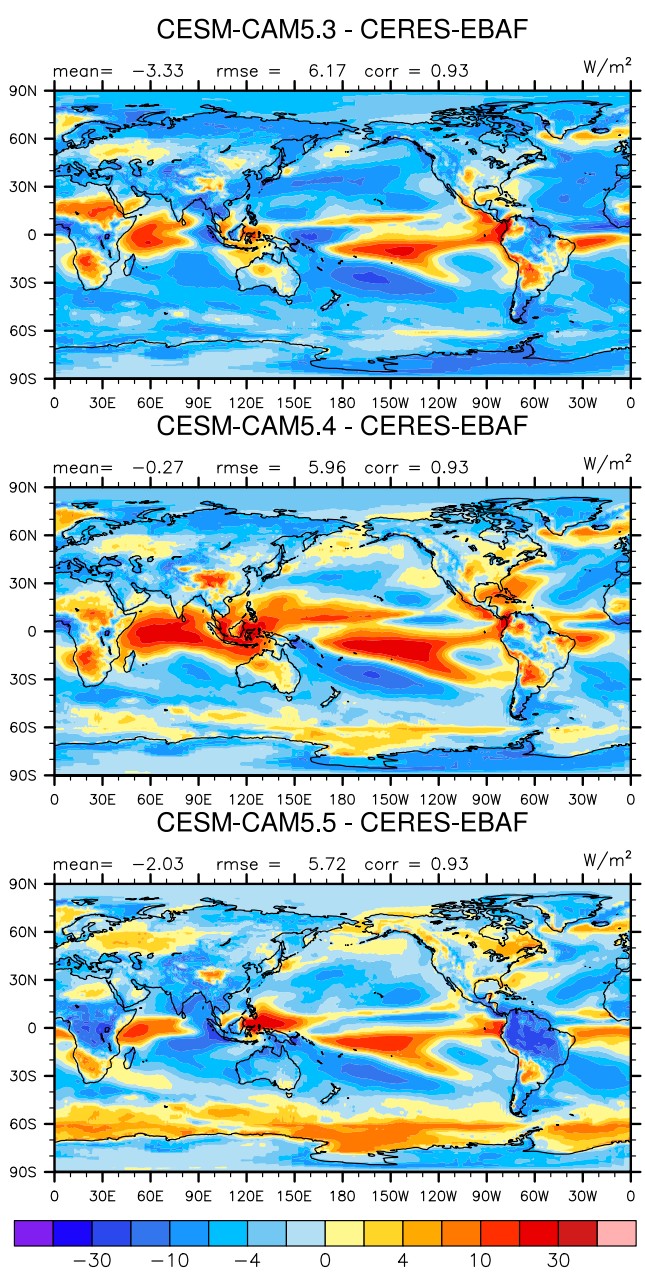

**Figure 3.** Same as Fig. 2 but for longwave cloud forcing biases.





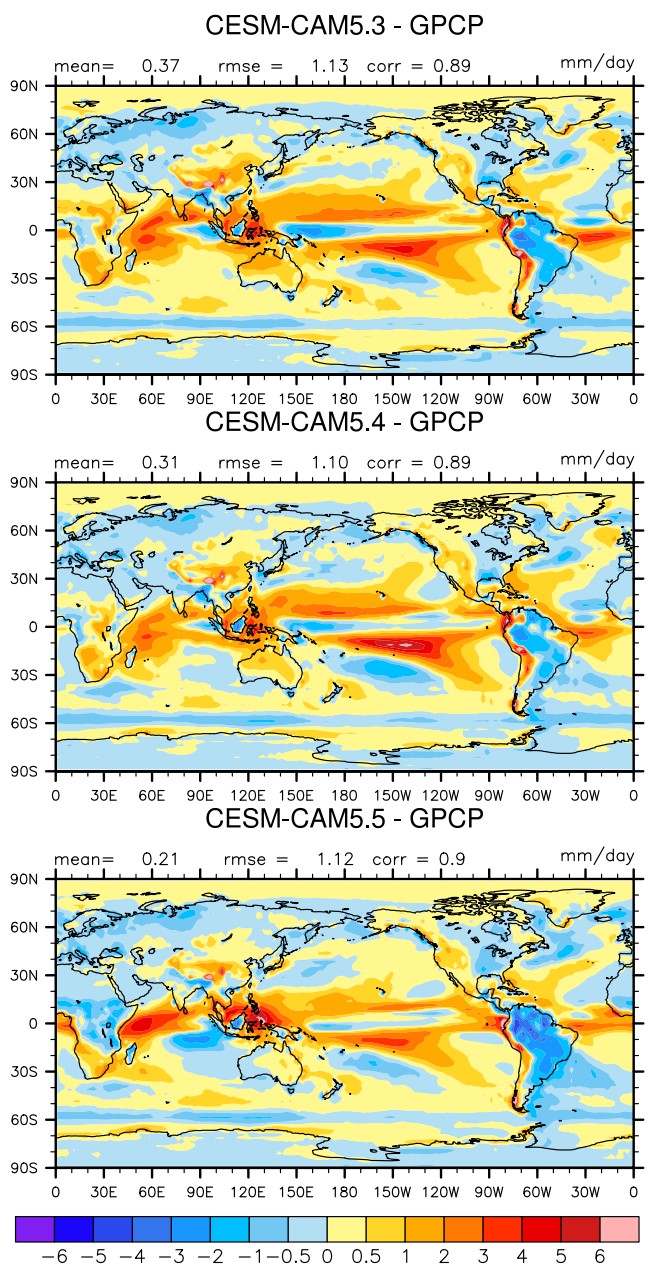

**Figure 4.** Same as Fig. 2 but for precipitation biases computed relative to GPCP observations.

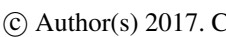



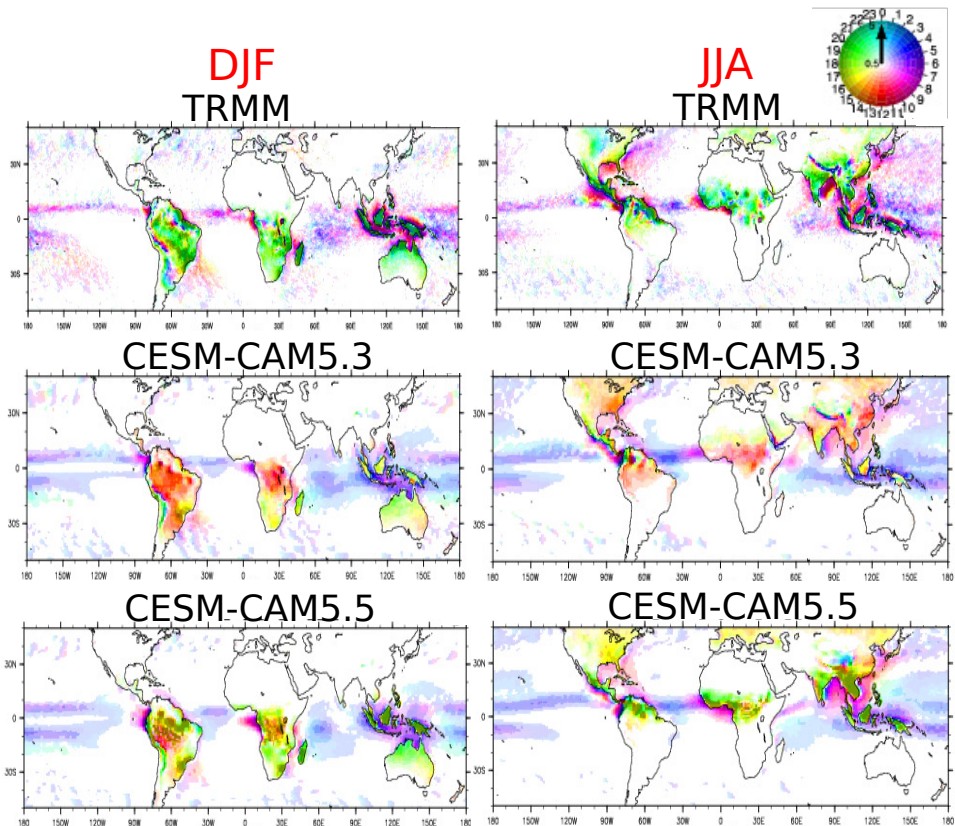

**Figure 5.** Diurnal cycle of precipitation maps for TRMM observations (top row), CESM-CAM5.3 (middle row), and CESM-CAM5.5 (bottom row) for December, January, February (DJF, left column) and June, July, August (JJA, right column). The color hue denotes the local time of day of the maximum precipitation rate, while the shading denotes the intensity of the precipitation.



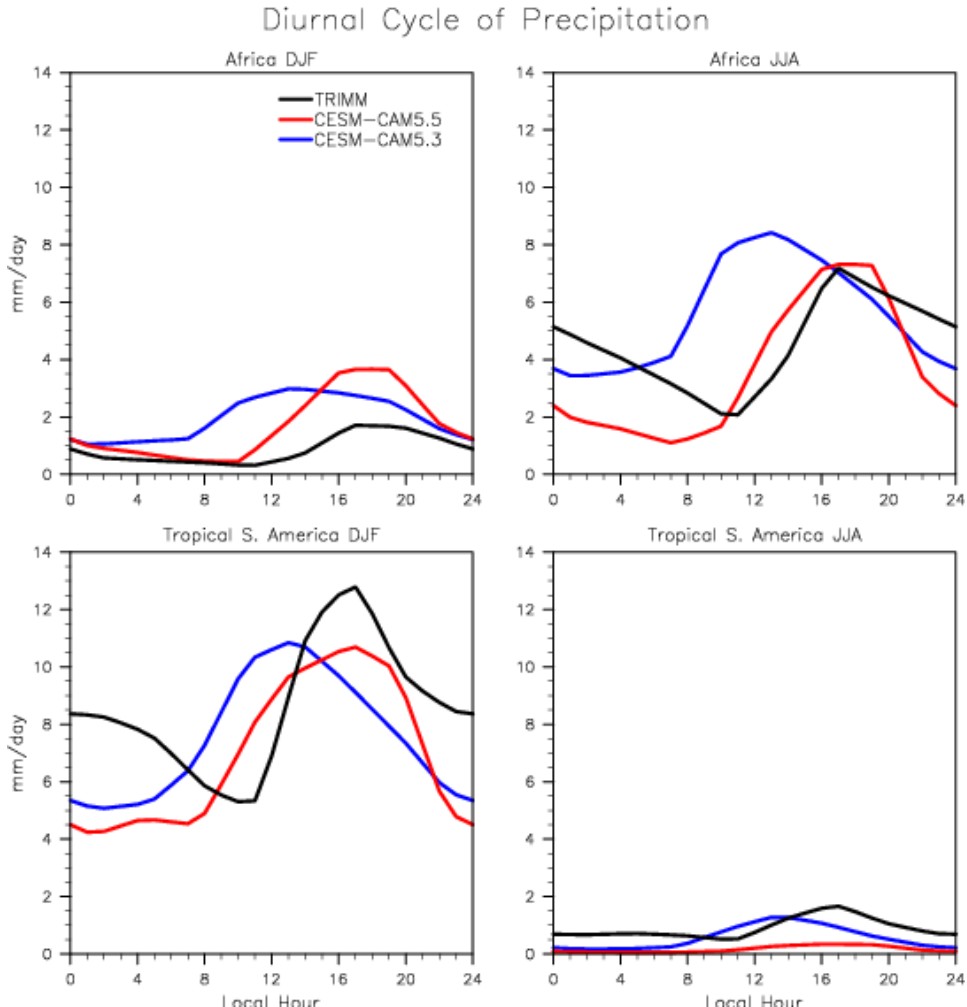

**Figure 6.** Timing composites of the diurnal cycle of precipitation for TRMM observations (black curves), CESM-CAM5.3 (blue curves), and CESM-CAM5.5 (red curves) for DJF (left column) and JJA (right column). Top row denotes the composites for an area average over tropical Africa (20° to 30° E and 0° to 10° N, while bottom row denotes the composites for an area average over tropical South America (65° to 80° W and 20° to 5° S.)





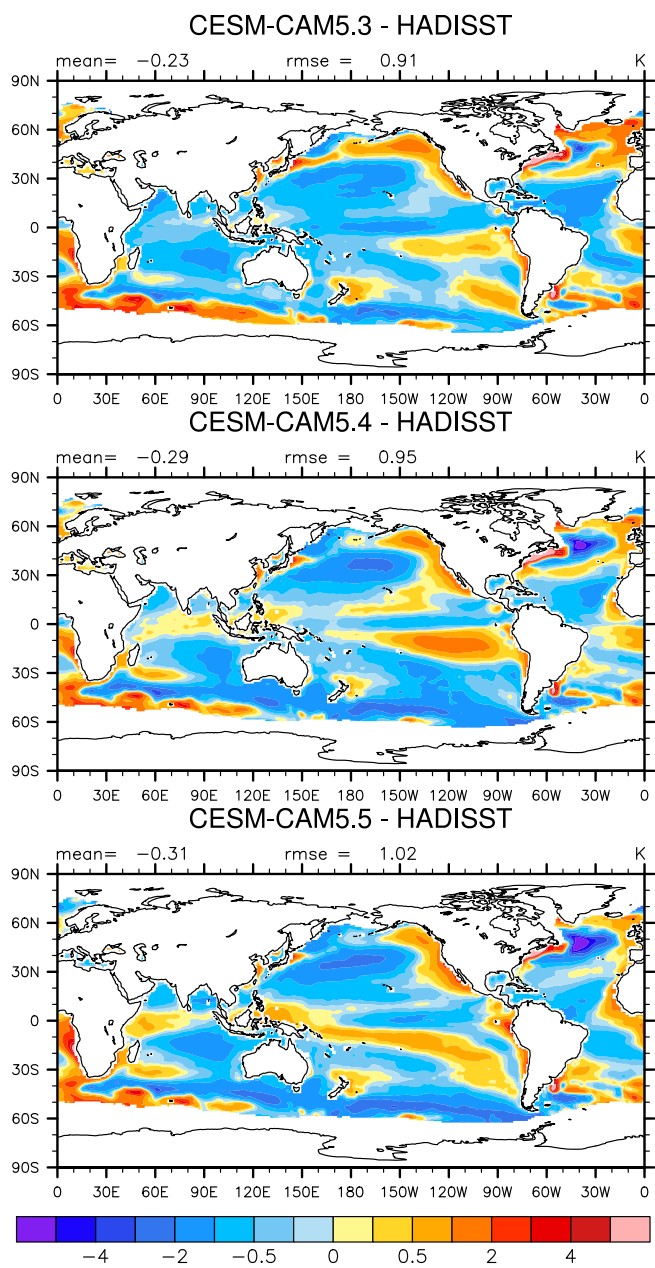

**Figure 7.** Same as Fig. 2 but for sea surface temperature biases computed relative to pre-industrial HadISST observations.





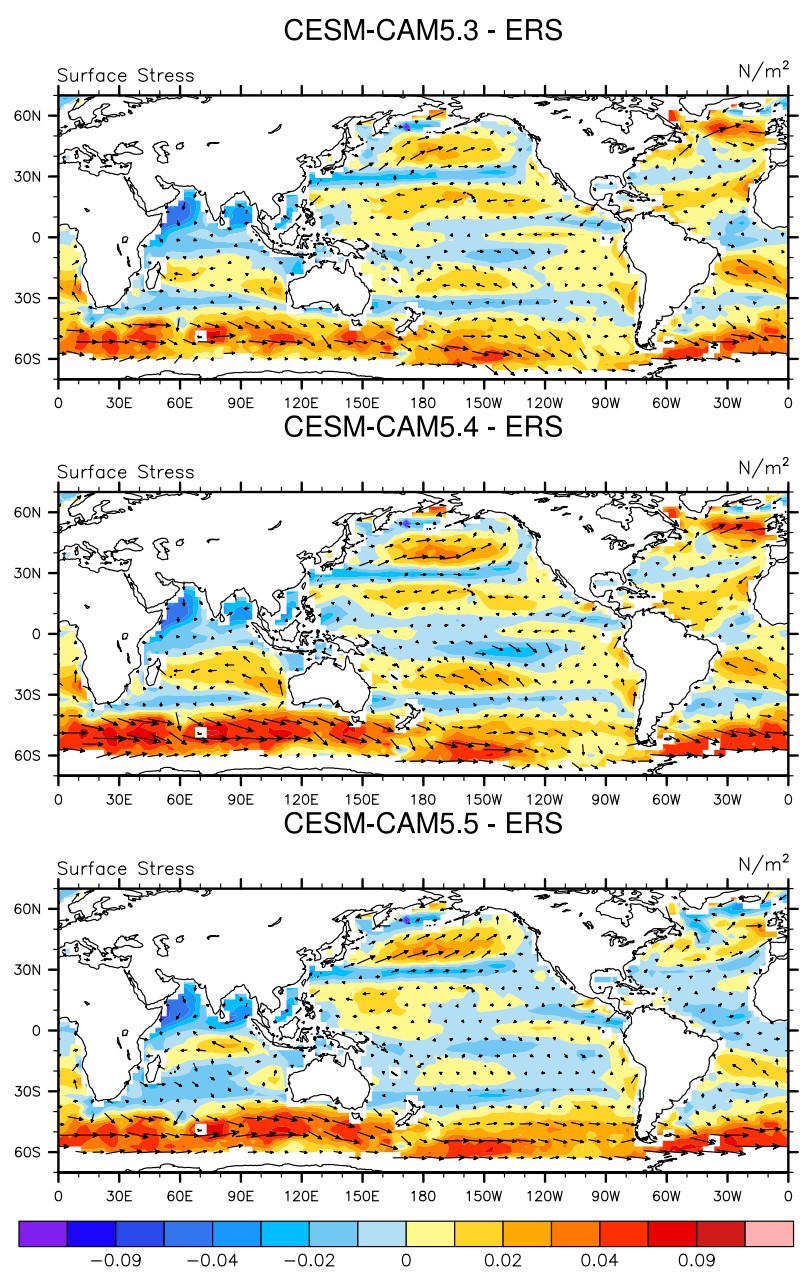

**Figure 8.** Same as Fig. 2 but for surface stress biases computed relative to ERS observations.





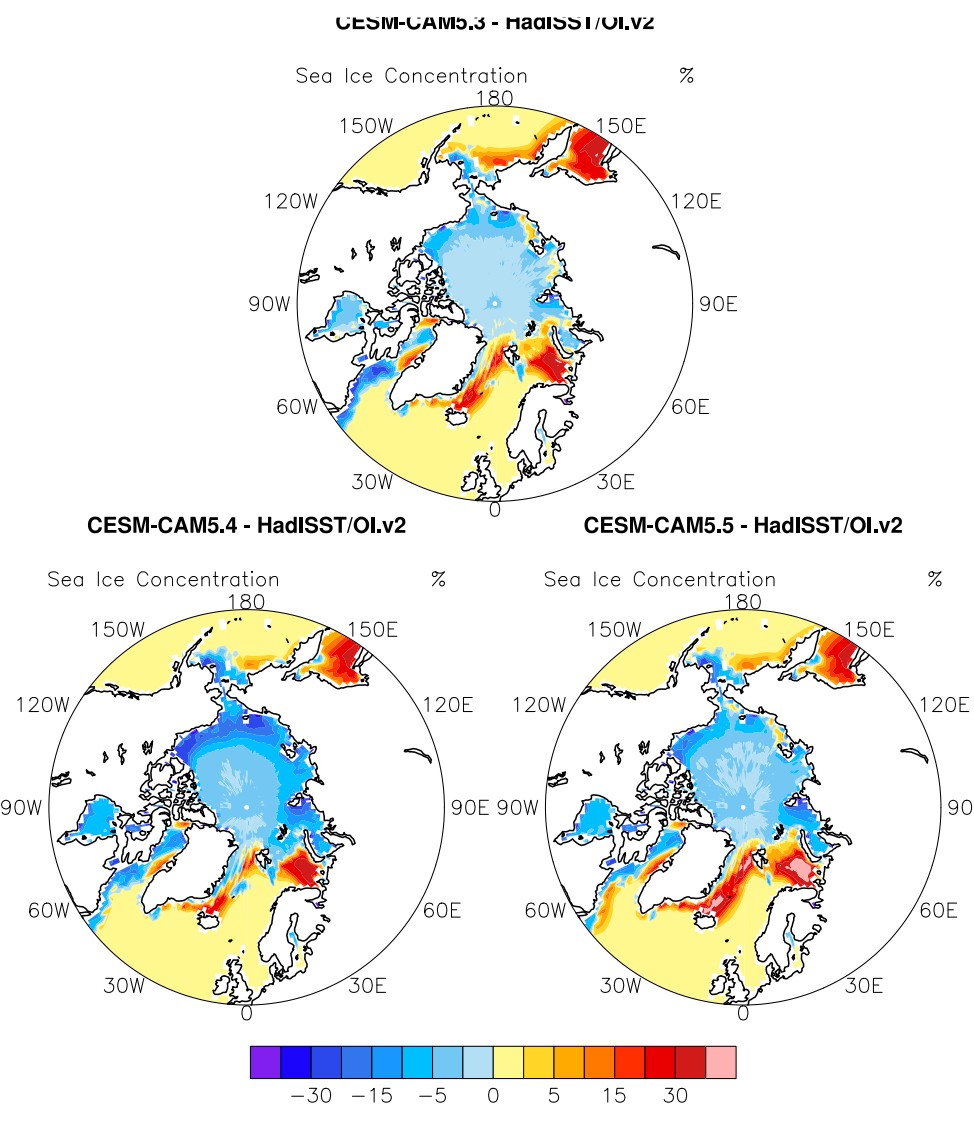

**Figure 9.** Same as Fig. 2 but for sea ice concentration over the north pole computed relative to pre-industrial HadISST observations.





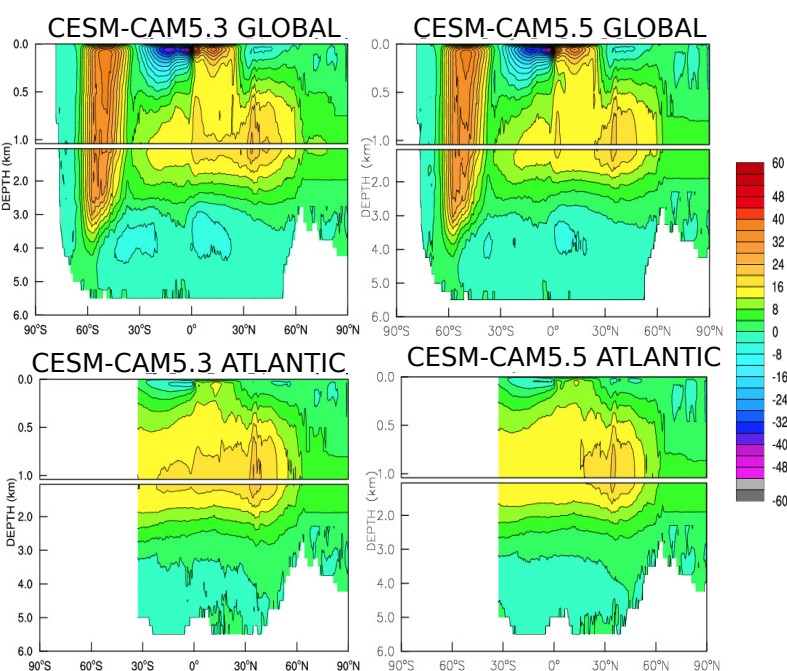

**Figure 10.** Global meridional overturning circulation (MOC, top row) and Atlantic MOC (bottom row) for CESM-CAM5.3 (left column) and CESM-CAM5.5 (right column). Units are Sv.





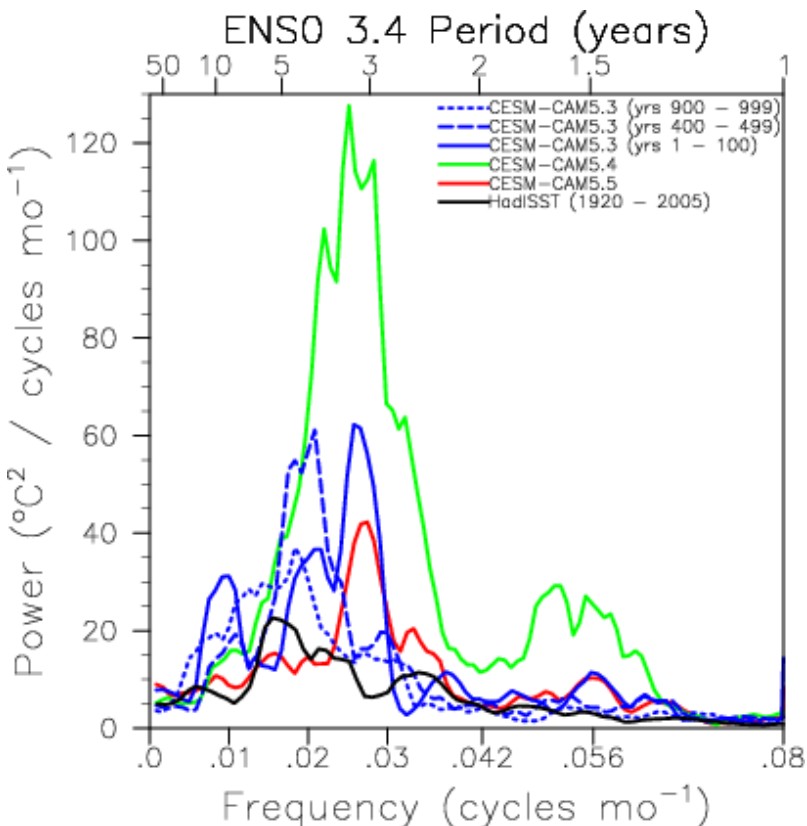

**Figure 11.** ENSO 3.4 power spectra for HadISST observations (black curve), CESM-CAM5.3 (blue curves), CESM-CAM5.4 (green curve), and CESM-CAM5.5 (red curve). Years 100 - 199 are displayed for CESM-CAM5.5 while the power spectra for CESM-CAM5.4 represents the entire simulation. For CESM-CAM5.3, the spectra is displayed for various 100 year samples of the run, as denoted by the legend. Note the range of scale on the y-axis varies between panels.





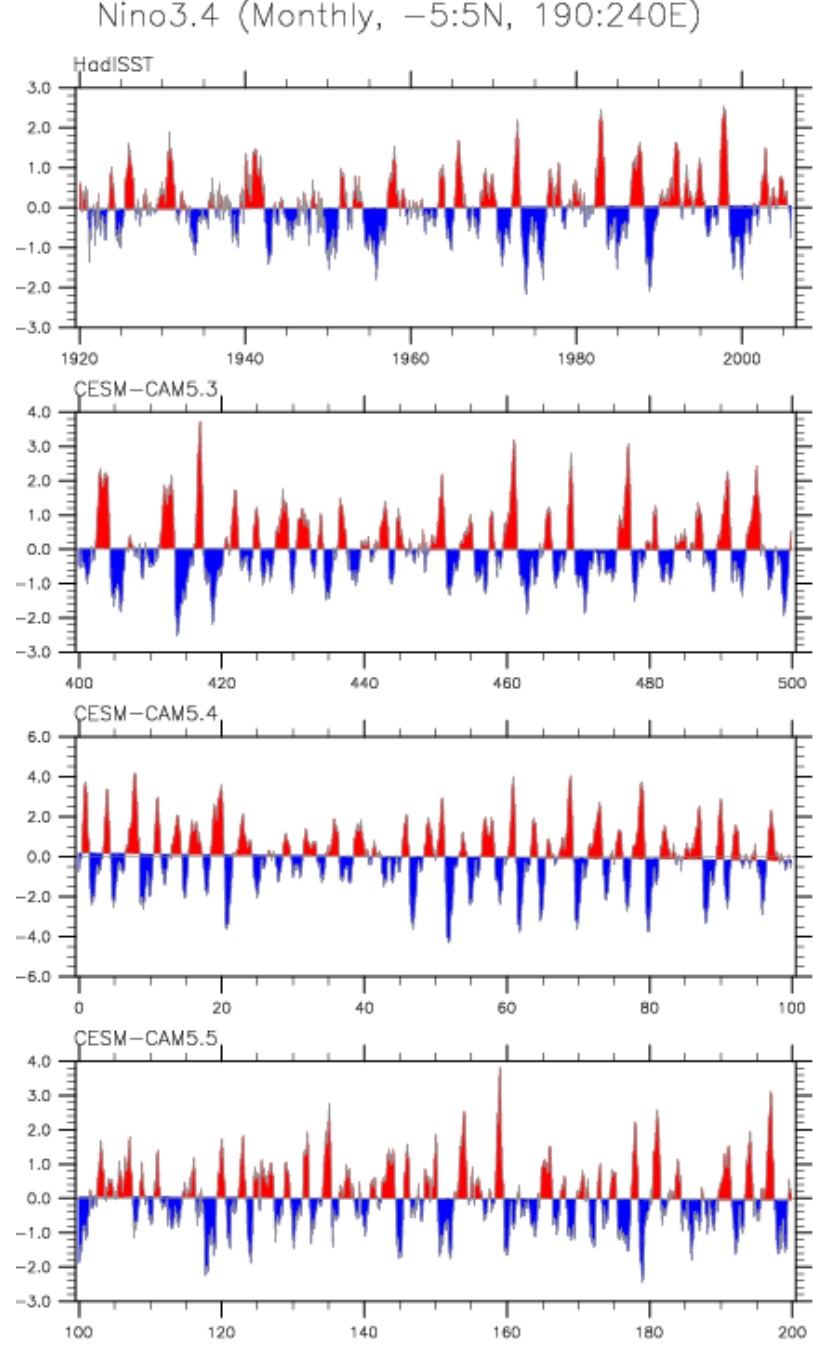

**Figure 12.** ENSO 3.4 time series for HadISST observations (top row), CESM-CAM5.3 (second row, years 400-499), CESM-CAM5.4 (third row, years 0-99), and CESM-CAM5.5 (bottom row, years 100-199).





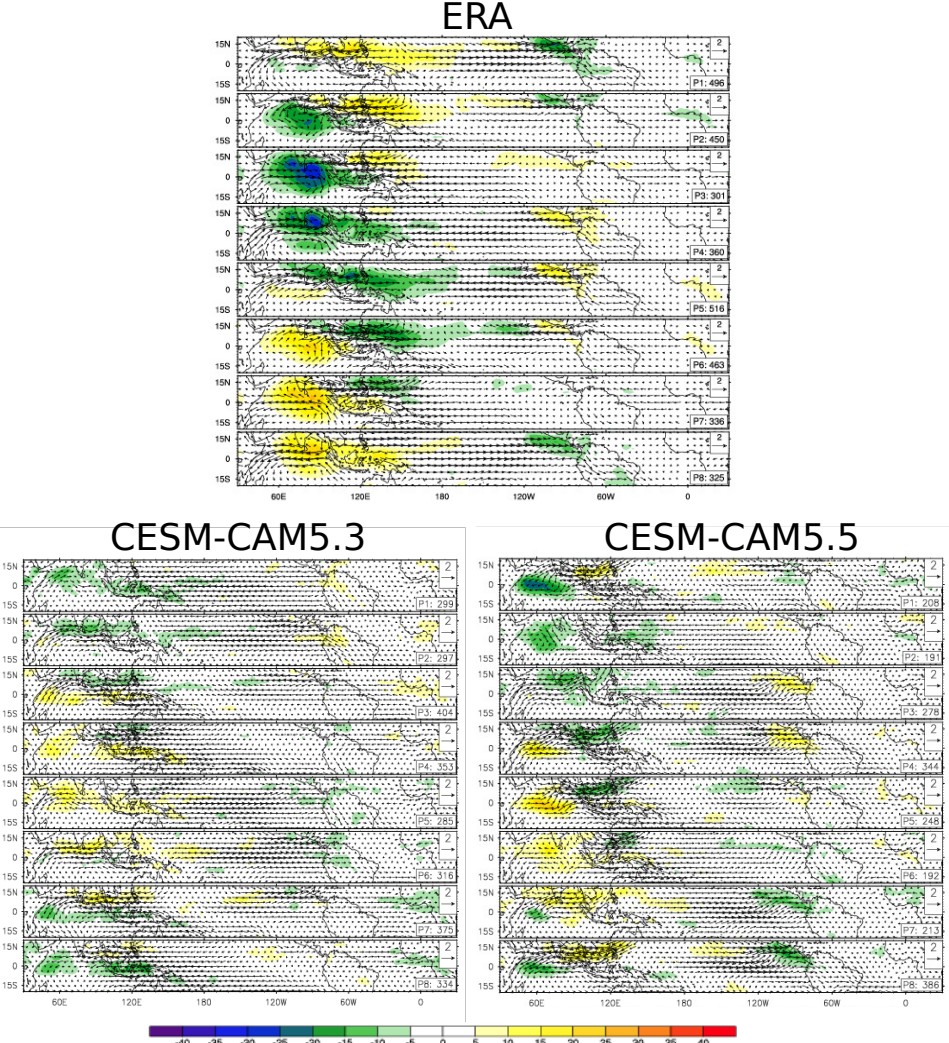

**Figure 13.** The composite life cycle of the 20 to 100 day bandpass-filtered daily anomalies of OLR (color) and wind vectors at 850 hPA during boreal winter (November through April) for ERA-Interim (top row), CESM-CAM5.3 (bottom row, left column), and CESM-CAM5.5 (bottom row, right column).





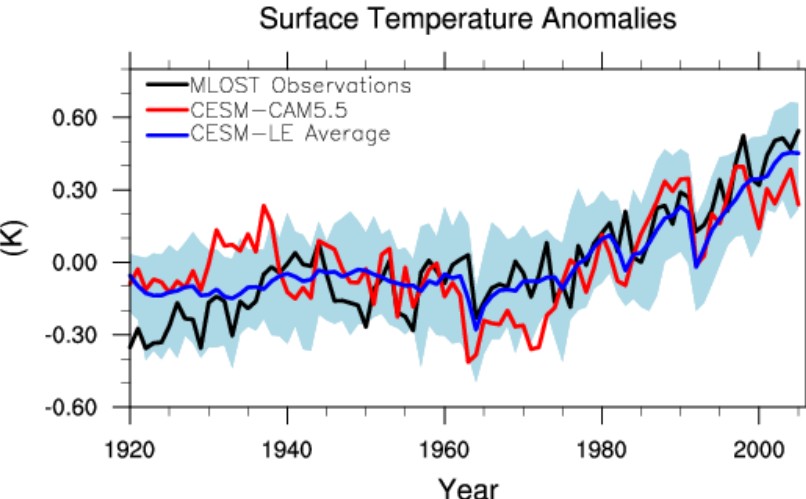

**Figure 14.** Evolution of the globally averaged surface temperature for the 1920-2005 period of the historical run for MLOST observations (black curve), CCSM4 ensemble average (green curve), CESM large ensemble average (blue curve), and CESM-CAM5.5 (red curve). The blue shading denotes the CESM large ensemble spread.



**Table 1.** Summary of physics used in each model version. Citation key: ZM1995 = Zhang and McFarlane (1995), PB2009 = Park and Bretherton (2009), BP2009 = Bretherton and Park (2009), P2014 = Park et al. (2014), MG1 = Morrison and Gettelman (2008), MG2 = Gettelman and Morrison (2014), G2010 = Gettelman et al. (2010), CLUBB = Golaz et al. (2002a), MAM3 = Liu et al. (2012), MAM4 = Liu et al. (2015), RRTMG = Iacono et al. (2008).

| Physics | CAM5.3 | CAM5.4 | CAM5.5 |
|---|---|---|---|
| Deep Convection | ZM1995 | ZM1995 | ZM1995 |
| Shallow Convection | PB2009 | PB2009 | CLUBB |
| PBL | BP2009 | BP2009 | CLUBB |
| Warm Cloud Macrophysics | P2014 | P2014 | CLUBB |
| Cold Cloud Macrophysics | G2010 | G2010 | G2010 |
| Microphysics | MG1 | MG2 | MG2 |
| Aerosol | MAM3 | MAM4 | MAM4 |
| Radiation | RRTMG | RRTMG | RRTMG |




**Table 2.** Summary of coupled simulations performed

| Model | Pre-industrial | 20th Century |
|---|---|---|
| CESM-CAM5.3 | 2100 years | 1850 to 2005: 1 member |
| | | 1920 to 2005: 37 members |
| CESM-CAM5.4 | 120 years | Not performed |
| CESM-CAM5.5 | 200 years | 1850 to 2005: 1 member |





**Table 3.** Summary of aerosol perturbation experiments. Values shown represent the differences in simulations using present day (2000) - pre-industrial (1850) aerosol emissions. All values are in W/m$^2$. All simulations are run with prescribed climatological SSTs where the only difference is in the aerosol emissions. RFP is defined as the difference between the top of model radiation imbalance for simulations with present day and pre-industrial aerosol emission. Values from the CAM5.3+MG2 simulation are from Gettelman et al. (2015). The CAM5.5+SB2001 simulation represents a configuration of CAM5.5 run with the Seifert and Beheng (2001) autoconversion and accretion physics.

| Simulation | RFP | $\Delta$SWCF | $\Delta$LWCF | AIE |
|:---:|:---:|:---:|:---:|:---:|
| CAM5.3 | -1.3 | -1.7 | +0.5 | -1.2 |
| CAM5.3+MG2 | -1.0 | -0.9 | +0.1 | -0.8 |
| CAM5.3+MG2+CLUBB | -1.6 | -1.2 | +0.0 | -1.2 |
| CAM5.4 | -1.6 | -1.2 | +0.1 | -1.1 |
| CAM5.5 | -1.8 | -1.8 | +0.4 | -1.4 |
| CAM5.5+SB2001 | -1.5 | -1.0 | +0.0 | -1.0 |