# Peer review of "The Path to CAM6: Coupled Simulations with CAM5.4 and CAM5.5"

_Geoscientific Model Development, 2017_

## Short Comment (SC1) · 12 Jul 2017

Dear authors,

the paths given in the code and data availability section are not readable. Please provide them in a readable form in an authors comment during the discussion phase and in the final revised version.

Best regards, Astrid Kerkweg

---

## Referee Comment (RC1) · Anonymous Referee #1 · 8 Aug 2017

This paper documents coupled simulations of two major developmental versions of Community Atmosphere Model (CAM) towards CAM6. Critical mean climate quantities and variabilities that are commonly used to characterize model performance are presented and discussed in an incremental manner designed to illustrate the impact of two set of major changes expected to be adopted by CAM6. The changes include new microphysics, aerosol, and ice nucleation in CAM5.4 configuration, and additionally an unified parameterization with an assumed-PDF (CLUBB) for turbulence, shallow convection and warm cloud macrophysics in CAM5.5 configuration. Important improvements are identified in each configuration, along with some degradations; and the attributions of them are convincing. Particularly this is for the first time the performance of the emerging assumed-PDF (CLUBB) method is documented in coupled mode, which

is expected to be a very useful reference for further development and application in CAM or other climate models. The presentation is succinct, thoughtful and well organized, along with many useful insights on tuning, coupling and perspective of CLUBB. The paper is suitable for GMD and can be published essentially in current form, after addressing the minor specific comments below.

Specific comments:

1. Only the atmosphere model is described in the section 2 for model description. Given that this work is to document the coupled simulations, it is useful to also briefly describe other model components used, assuming the same are used for all the CAM configurations in this work.

2. Figure 1 on the preindustrial runs: CESM-CAM5.3 at year 402 is assumed to have a globally averaged surface temperature near the stable equilibrium of 287.0K. Why CESM-CAM5.4 and CESM-CAM5.5, which were initialized with CESM-CAM5.3 at year 402, have substantially higher initial global mean surface temperature? This appears inconsistent given the description in the text. Is there something missing?

3. Page 8 line 4, it is not an accurate statement suggesting that "improvements stem from reduction in magnitude of the errors", given pattern correlation coefficient remain unchanged. From Figure 3, it can also be seen that both error magnitudes and patterns change; and there exist quite regions with error magnitudes become larger.

4. Figure 10 on relative AMOC strength between CESM-CAM5.3 and CESM-CAM5.5: the authors speculated that the difference in simulated surface wind stress in the north Atlantic could be the likely cause. Large difference in southern mid-latitude surface wind stress between them could be an even larger factor (e.g., Delworth and Zeng 2008, GRL, doi:10.1029/2008GL035166). Suggest to review and revise this speculative attribution.

5. Figure 6 includes the diurnal composite of precipitation for the tropical Africa, but

essential no description in text. Suggest to add some description for it, though the points to make can largely be reflected in the Amazon composites.

6. Page 5 line 21, redundant word "that" is used.

7. Page 11, last line, given the context, "inter-annual seasonal tropical variability" should be "intra-seasonal . . .".

---

## Referee Comment (RC2) · Anonymous Referee #2 · 16 Sep 2017

The manuscript documents two versions of CAM, each a step towards the development of CAM6. The first one has new microphysics, aerosol, and ice nucleation changes, the second one implements pdf-based unified cloud parameterization to replace the turbulence, shallow convection, and warm cloud macrophysics. The newest model has lower cloud forcing bias, better nino3.4, better precip diurnal cycle, subtropical wind stress and less of a double ITCZ. It also has worse Amazon precip, colder SSTs in general, and sea ice that is too thin. The 20th century simulation ends up colder in new model (too cold), due to stronger aerosol indirect, and the authors propose a fix. The authors conclude that model is ready to integrate into NCAR's next generation model.

The documentation of changes in model simulation during the development stages of the CMIP6 model, along with the attribution of these changes to specific elements

of the coupled model, is clearly an important contribution to understanding climate change processes and our models' ability to simulate them. This manuscript requires the major revisions outlined below in the general and detailed comments in order to be ready for publication in GMD.

General Comments ─────────

1) The issue of whether the 5.4 and 5.5 simulations were tuned: The authors state throughout the manuscript that the 5.4 and 5.5 models were tuned for radiative balance but not for SST, and so improvements are due to the former and degradations are due to the latter. The authors need to explain why an untuned version of the model is ready for publication.

2) Attribution claims are unwarranted. The entire moist physics and cloud forcing was replaced from 5.3 to 5.4, say, and the text speculates about which change in the model was responsible for which change in the behavior. examples are: i) p8 line 26: "can be attributed to interactions of CLUBB with the ZM scheme and/or feedbacks to/from the coupled system with CAM5.5" ii) p9 line 1 : "the difference in precipitation simulation is likely linked to differences in parameterized physics as opposed to biases in the large-scale circulation"

3) Statements about differences among different model version results are qualitative in nature in many instances ("subtantially", "much too....") and are not put in the context of internal variability or significance. This makes it difficult to understand the changes.

4) Section 4.5 is an outstanding example of how to report on and discuss attribution. The other sections of the manuscript could benefit from this type of discussion or reporting of results of sensitivity experiments.

Detailed comments ─────

p3 - discussion of implementations of CLUBB into global models. AM3 wrote the code to do it and tested it, but did not adapt and will not use in CMIP runs. please make this

clear.

page 5 line 24 and then line 31 says 5.5 tuning.... abstract says not tuned. is it or not? [if not, this paper not ready yet for evaluation]

page 5 line 27 - autoconv of ice to snow increased to increase cloud? seems counter-intuitive on the face of it. or does increasing Dcs mean decreasing autoconversion? if so, please make this clear.

page 6 line 1 - also needs a few more words - why suppressing turb mixing tends to increase low cloud? won't mixing more moisture for instance tend to increase low cloud (ie., Lock's coupled BL idea)?

page 6 parag with line 3 - why insist that the tuning was only partial? please remove this discussion here and elsewhere. again, if the tuning was only partial the model is not ready for publication.

p7 l6 - is 5.5 equillibrated? looks like still rising.

p7 l15 - the periods chosen for 5.4 and 5.5 seem to be periods when global sfc temp is rising, almost as fast as it is falling in the earlier period. also - is it the author's assertion that the "offset" from pre-industrial to present day (period of simulation vs period of validation data) does not have geographical structure? please elaborate a bit more about the offset.

p7 line 18 - please discuss significance of the differences in RMSE and pattern correlation. is 0.94 really an improvement over 0.93?

p7 line 21 - i see a southern ocean difference in 5.3 of approximately +15 or so w/m***2 and in 5.4 of approximately -15 w/m**2. certainly a difference in behavior but not a "significant reduction" of bias. please correct this statement, or please quantify the zonal mean difference, say and report on a significance test.

p7 end of discussion of SWCF - the authors comment on the transition from 5.4 to 5.5

Interactive
comment

without commenting on what looks like a degradation in the global mean difference.

p8 l2-3 - please remove or restate this sweeping assessment. as discussed below, it is not clear that each model is an improvement over the previous.

p8 l9 - is this speculation about the autoconversion? if so, please state that it is probably or possibly. if not, please report on experiments that were done to isolate the impact of this parameter, and their applicability to the simulations reported here.

p8 l13 - which metric is being referred to as skill score? please rephrase this to refer to all or one of the metrics provided in the figure.

p8 l31 - which panel is this sentence referring to? 5.5? please rephrase this sentence.

p9 l2 - 5.5 also seems to have degraded the precip over the maritime continent.

p9 l14 - the authors state that the reason for the improved diurnal cycle "appears to be" due to CLUBB - please either show or report on results of atmosphere only (or coupled, if you have them) sensitivity studies demonstrating this. especially since this point is mentioned again (p9 l 23).

p9 l18 - summer precip in JJA in Amazon seems to have almost gone away in JJA - any speculation? is this an important result?

p9 l29 - please say mean error and rmse instead of skill score. also - are these "successive degradations" significant statistically?

p9-10, discussion of figure 7 - There are several issues with this discussion that warrant addressing by the authors. a) why would the "but it wasn't tuned" impact the SST and not the precip, for instance? b) line 29 p9 says "worsening", but line 33 says "not substantially worse". is it worse (statistically) or is it not? please clarify. c) the statement in line 33 of p9 that the cloud forcing can be adjusted to fix SST bias is in stark contrast to the statement on p10 line 3 that states that 5.4 cloud forcing is better but SST is worse, perhaps due to compensating biases in the ocean.

the entire discussion of the attribution of (perhaps) SST errors that increase with model advance is vague and speculative. please re-purpose this discussion. either show that a re-consideration of the tuning will improve the SST and that the eventual CMIP 6 model will address this, or wait until those results are in to publish the manuscript.

p10 l12 - please explain why the stress errors cause SST errors and not the other way around. or state that they are related. also please quantify the difference (the "degradation" to 5.4 and the "improvement" to 5.5 in the subtropics).

p10 l19-27 - same issue as with the SST. i see one region of "degradation", and the statements about fixing it with tuning need to be reconsidered.

p10 l27 - this is not an attribution of the error, but speculation in a broad sense. please either show some sensitivity results or remove the statement.

p10 l30 - please provide an observational estimate for comparison. either include an extra set of panels in the figure or provide the number at 35N, 1 km depth. please also provide an estimate of variability - is 23 Sv different from 26?

p11 l1 - please remove the statement about the likely cause for the change in the AMOC. there is no basis provided for this. the authors could be correct, or it could be a myriad of reasons, including some offered by the authors when discussing SST or wind stress differences.

p11 l29 - fig 12 adds little to this discussion. please consider removing it.

p12 l3 - the lifecycle phases are not mentioned or indicated in the figure. please either amend the figure or remove the mention of the lifecycle phases from the text.

p12 l10 - studies have shown that the addition of a (better) shallow conv parameterization increased the amplitude of MJO variability in atmos only simulations. please refer to one of these. shouldn't CLUBB be expected to help in this regard? what do atmos only simulations show the impact of CLUBB to be?

p12 l23 - syntax "a couple exceptions"

p13 l4,11 repeat the same information

p14 l20 this statement (each one improved over the other) was also stated in the results section but was not clearly borne out by the figure or the metrics reported. if the results section includes some discussion of variability or significance this statement would be appropriate.

p14 l21 please rephrase "big"

p14 l24 - please modify discussion of tuning. it is confusing to state that 5.5 and 5.4 were tuned for radiation but not for SST, so its not surprising that they are worse, but they aren't that much worse. in the text it is suggested that the degradation of SST is due to cloud forcing. please modify this entire discussion. and please discuss why an untuned model is ready for reporting in the literature.

---

## Author Comment (AC1) · 31 Oct 2017

This has been fixed in the revised document. The model code used for this paper may be accessed at CESM developers repository at: https://svn-ccsm-models.cgd.ucar.edu/cam1/branch_tags/cam55_reproduce_tags/cesm1_4_beta01_n27_cam5_3_77
.

Climatology files of the model runs used to generate figures in this paper have been published at zenodo.com (doi:10.5281/zenodo.815593).
* * *

---

## Author Comment (AC2) · 31 Oct 2017

We would like to thank anonymous referee #1 for volunteering their time to review this manuscript and for offering suggestions to make it better. Please see responses to each specific comment below:

1) Only the atmosphere model is described in the section 2 for model description. Given that this work is to document the coupled simulations, it is useful to also briefly describe other model components used, assuming the same are used for all the CAM configurations in this work.

Author response (AR): A new sub-section has been added entitled "Component Models" to address this.

[Figure]

2) Figure 1 on the preindustrial runs: CESM-CAM5.3 at year 402 is assumed to have a globally average surface temperature near the stable equilibrium of 287.0K. Why CESM-CAM5.4 and CESM-CAM5.5, which were initialized with CESM-CAM5.3 at year 402, have substantially higher global mean surface temperature? This appears inconsistent given the description in the text. Is there something missing?

AR: The initialization of CESM-CAM5.4 and CESM-CAM5.5 means the ocean state is the same as CESM-CAM5.3 as they start. Since CESM-CAM5.4 and CESM-CAM5.5 have a different atmosphere, and a slightly different energy budget, they will reach a different equilibrium temperature.

3) Page 8 line 4, it is not an accurate statement suggesting that "improvements stem from reduction in magnitude of the errors", given pattern correlation coefficient remain unchanged. From Figure 3, it can also be seen that both error magnitudes and patters change; and there exist quite regions with error magnitudes become larger.

AR: This statement has been removed and the discussion modified accordingly.

4) Figure 10 on relative AMOC strength between CESM-CAM5.3 and CESM-CAM5.5: the authors speculated that the difference in simulated surface wind stress in the north Atlantic could be the likely cause. Large difference in southern mid-latitude surface wind stress between them could be an even larger factor (e.g., Delworth and Zeng 2008). Suggest to review and revise this speculation attribution.

AR: The Delworth and Zeng 2008 paper has been cited and this has been mentioned as a possible reason for the differences in the simulated AMOC between the model configurations.

5) Figure 6 includes the diurnal composite of precipitation for the tropical Africa, but essential no description in text. Suggest to add some description for it, though the points to make can largely be reflected in the Amazon composites.

AR: Thanks for pointing out this oversight. A short discussion has been added for the

African diurnal cycle of precipitation.

6) Page 5 line 21, redundant word "that" is used.

AR: Fixed

7) Page 11, last line, given the context, "inter-annual seasonal tropical variability should be "intra-seasonal . . .".

AR: Fixed

---

## Author Comment (AC3) · 31 Oct 2017

Response to Anonymous Referee #2:

We would like to thank anonymous referee #2 for volunteering their time to review this manuscript and for offering suggestions to make it better. Please see responses to each specific comment below:

General Comments

1) The issue of whether the 5.4 and 5.5 simulations were tuned: The authors state throughout the manuscript that the 5.4 and 5.5 models were tuned for radiative balance but not for SST, and so improvements are due to the former and degradations are due to the latter. The authors need to explain why an untuned version of the model is ready

for publication.

Author response (AR): Most of the statements blaming degradations on an untuned model have been removed, replaced, or very much understated in the in the revised manuscript. The statement that the CAM5.4 and CAM5.5 models were only tuned at this point for radiative balance and stable pre-industrial runs in the model descriptions has been retained.

With that said, we would like to clarify CAM5.4 and CAM5.5 were tuned when under development in atmosphere only simulations (thus, these are not strictly untuned models). However, these tunings were not modified for coupled simulations except to address radiation balance and/or stability issues. This has been noted explicitly in the document.

We argue that these models are ready for publication to serve as a documentation of the changes that occur in the simulated climate when progressive changes are made to the CESM atmosphere model. Aggressive tuning was not performed in these simulations to save on computational resources since the models would have to be retuned anyway once the newer component models and final CMIP forcing data is introduced. We feel these simulations serve as valuable waypoints for such a widely used community model, to understand the effects of new parameterizations on the climate absent a heavy retuning of the model.

2) Attribution claims are unwarranted. The entire moist physics and cloud forcing was replaced from 5.3 to 5.4, and the text speculates about which change in the model was responsible for which change in the behavior. Examples are: i) p8 line 26: "can be attributed to interactions of CLUBB with the ZM scheme and/or feedbacks to/from the coupled system with CAM5.5" ii) p9 line 1: "the difference in precipitation simulation is likely linked to differences in parameterized physics as opposed to biases in the large-scale circulation"

AR: Many of these attribution claims are removed or replaced with more speculative

comments.

3) Statements about differences among different model version results are qualitative in nature in many instances ("substantially", "much too. . .") and are not put in the context of internal variability or significance. This makes it difficult to understand the changes.

AR: These statements have been removed and/or modified and replaced with more quantitative statements.

4) Section 4.5 is an outstanding example of how to report on and discuss attribution. The other sections of the manuscript could benefit from this type of discussion or reporting of the results of sensitivity experiments.

AR: Reports on sensitivity studies have been mentioned where available.

Detailed Comments

1) P3 – discussion of implementations of CLUBB into global models. AM3 wrote the code to do it and test it, but did not adapt and will not use in CMIP runs. Please make this clear

Author response (AR): This has been noted.

2) Page 5 line 24 and then line 31 says 5.5 tuning. . . abstract says not tuned. Is it or not? [if not, this paper not ready yet for evaluation]

AR: This discrepancy has been corrected. Mention of tuning has been removed in abstract. We clarify in Atmosphere Model descriptions that CAM5.4 and CAM5.5 were tuned extensively in their atmosphere only configurations, but tuned in coupled mode to achieve radiation balance and a stable pre-industrial control run.

3) Page 5 line 27 – autoconv of ice to snow increased to increase cloud? Seems counterintuitive on the face of it. Or does increasing DCS mean decreasing autoconversion? If so, please make this clear.

AR: Meant to say that the threshold at which autoconversion occurs has been increased. This has been fixed in the text.

4) Page 6 line 1 – also needs a few more words – why suppressing turb mixing tends to increase low cloud? Won't mixing more moisture for instance tend to increase low cloud (i.e. Lock's coupled BL idea)?

AR: The gamma_coef parameter was poorly described in the original text. Have removed this comment and replaced with "decreasing the gamma parameter helps to decrease the skewness of vertical velocity and scalars, making the layer less cumuliform and more stratiform, with increased low-cloud cover"

5) Page 6 parag with line 3 – why insist that the tuning was only partial? Please remove this discussion here and elsewhere. Again, if the tuning was only partial the model is not ready for publication.

AR: Statements about tuning being partial have largely been removed and understated in regards towards explaining the results.

6) P7 l6 – is 5.5 equillibrated? Looks like still rising

AR: Have revised the statement to read that CAM5.5 appears to achieve a reasonable equilibrium. It is possible with a longer control run that the simulation could equilibrate at a slightly higher temperature.

7) P7 l15 – the periods chosen for 5.4 and 5.5 seem to be periods when global sfc temp is rising, almost as fast as it is falling in the earlier period. Also – is it the author's assertion that the "offset" from pre-industrial to present day (period of simulation vs period of validation date) does not have geographical structure? Please elaborate a bit more about the offset.

AR: We have checked (and noted in the document) that the results in this paper do not depend on the averaging period, as long as this averaging period occurs after reasonable equilibrium is achieved (for CAM5.5 after year 100, for CAM5.4 after year

75).

Present day cloud forcing will differ from pre-industrial due to (1) cloud response to surface temperature changes between 1850 and present (cloud feedbacks) and (2) anthropogenic forcing from aerosol cloud interactions. Both are on the order of 0.5 to 1.0 Wm-2. The first (cloud feedbacks) is thought to be positive, and the second (cloud forcing from aerosols) is thought to be negative, concentrated in the N. Hemisphere. This has been noted in the revised manuscript.

8) P7 line 18 – please discuss significance of the differences in RMSE and pattern correlation. Is 0.94 really an improvement over 0.93?

AR: Downplayed. We simply note that CAM5.4 and CAM5.5 improves the pattern correlation score over CAM5.3 but do not state that CAM5.5 is improved over CAM5.4 in this regard.

9) P7 line 21 – I see a southern ocean difference in 5.3 of approximately +15 or so w/m**2 and in 5.4 of approximately -15 w/m**2. Certainly a difference in behavior but not a "significant reduction" of bias. Please correct this statement, or please quantify the zonal mean difference, say and report on a significance test.

AR: This is correct. The improvement in bias is coming along the 60S transect for CAM5.4 and CAM5.5, while ~40S there is a now a positive SWCF radiative bias that is not necessarily an improvement over CAM5.3. This has been quantified in the text. In addition, we added figure 3 (zonal averages and differences from observations for cloud forcing) to better demonstrate this.

10) P7 end discussion of SWCF – the authors comment on the transition from 5.4 to 5.5 without commenting on what looks like a degradation in the global mean difference.

AR: We are unclear about this comment. CAM5.5 has the lowest global mean difference to observations (-0.83 W/m2), thus appears to be an improvement over CAM5.4 and CAM5.5.

11) P8 l2-3 – please remove or restate this sweeping assessment. As discussed below, it is not clear that each model is an improvement over the previous.

AR: This statement has been modified and very much downplayed.

12) P8 l9 – is this speculation about the autoconversion? If so, please state that it is probably or possibily. If not, please report on experiments that were done to isolate the impact of this parameter, and their applicability to the simulations reported here.

AR: It was noted that experiments were done to isolate the responsibility of this bias to tuning. Personal communication with Cecile Hannay was noted on this.

13) P8 l13 – which metric is being referred to as skill score? Please rephrase this to refer to all or one of the metrics provided in the figure.

AR: This statement has been cleared up.

14) P8 l31 – which panel is this sentence referring to? 5.5? please rephrase this sentence.

AR: It has been clarified that we are referring to CAM5.5 in this statement.

15) P9 l2 – 5.5 also seems to have degraded the precip over the maritime continent.

AR: This has been mentioned.

16) P9 l14 – the authors state that the reason for the improved diurnal cycle "appears to be" due to CLUBB – please either show or report on results of atmosphere only (or coupled, if you have them) sensitivity studies demonstrating this. Especially since this point is mentioned again.

AR: We have reported our experiences that this diurnal cycle result is a very robust feature in every simulation we have performed with CLUBB where this has been investigated (even going back to the first CAM-CLUBB simulations performed in Bogenschutz et al. 2013).

17) P9 l18 – summer precip in JJA in Amazon seems to have almost gone away in JJA – any speculation? Is this an important result?

AR: This has been noted in the document as a potential important source of improvement to the mean state climatological precip field. At this time, there is no speculation given.

18) P9 l29 – please say mean error and rmse instead of skill score. Also – are these "successive degradations" significant statistically?

AR: We have rephrased the wording here. It appears that the change in RMSE error from CAM5.3 to CAM5.5 is statistically significant at 95% confidence and this has been noted.

19) P9-10, discussion of figure 7 – There are several issues with this discussion that warrant addressing by the authors. A) why would the "but it wasn't tuned" impact the SST and not the precip, for instance? B) line 29 p 9 says "worsening", but line 33 says "not substantially worse". Is it worse (statistically) or is it not? Please clafiy. c) the statement in line 33 of p9 that the cloud forcing can be adjusted to fix SST bias is in stark constrast to the statement on p10 line 3 that states that 5.4 cloud forcing is better but SST is worse, perhaps due to compensating biases in the ocean. The entire discussion of the attribution of (perhaps) SST errors that increase with model advance is vague and speculative. Please re-purpose this discussion. Either show that a re-consideration of the tuning will improve the SST and that the eventual CMIP 6 model will address this, or wait until those results are in to publish the manuscript.

AR: This discussion has been repurposed, with the discussion on tuning very much understated. Cold biases in the mid-latitudes are explained by the strong SWCF biases that now appear centered around ∼40S for CAM5.4 and CAM5.5.

20) P10 l12 – please explain why the stress errors cause SST errors and not the other way around. Or state that they are related. Also please quantify the difference (the

"degradation" to 5.4 and the "improvement" to 5.5 in the subtropics)

AR: This statement was removed after examining that wind stress biases are similar in atmosphere only configurations for CAM5.4 and CAM5.5 (and this is noted). Quantification for the degradation and improvement is noted.

21) P10 l19-27 – same issue as with SST. I see one region of "degradation", and the statements about fixing it with tuning need to be reconsidered.

AR: Statement about fixing it with tuning has been removed.

22) P10 l27 – this is not an attribution of the error, but speculation in a broad sense. Please either show some sensitivity results or remove the statement.

AR: This has statement has been revised to a speculation.

23) P10 l30 – please provide an observational estimate for comparison. Either include an extra set of panels in the figure or provide the number at 35N, 1 km depth. Please also provide an estimate of variability – is 23 Sv different from 26?

AR: References to an observational estimate have been reported in the text. It appears that all flavors of CESM are within the observational uncertainty.

24) P11 l1 – please remove the statement about the likely cause for the change in the AMOC. There is no basis provided for this. The authors could be correct, or it could be a myriad of reasons, including some offered by the authors when discussing SST or wind stress differences.

AR: This has been modified to read as a speculation. Other speculations for differences in the simulation are stated with cited literature.

25) P11 l29 – fig 12 adds little to this discussion. Please consider removing it.

AR: We favor retaining the figure to demonstrate the fairly reasonable La Nina periods that often follow El Nino events.

26) P12 l3 – the lifecycle phases are not mentioned or indicated in the figure. Please either amend the figure or remove the mention of the lifecycle phases from the text.

AR: This text has been modified.

27) P12 l10 – studies have shown that the addition of a (better) shallow conv parameterization increased the amplitude of MJO variability in atmos only simulations. Please refer to one of these. Shouldn't CLUBB be expected to help in this regard? What do atmos only simulations show the impact of CLUBB to be?

AR: Citations to some of these works have been added. We note both in the MJO discussion and conclusions that the merely modest improvements seen in the MJO with CLUBB are counter to these works.

28) P12 l23 – syntax "a couple exceptions"

AR: Fixed

29) P13 l4,11 – repeat the same information

AR: Repeated information removed

30) P14, l20 – this statement (each one improved over the other) was also stated in the results section but was not clearly borne out by the figure or the metrics reported. If the results section includes some discussion of variability or significance this statement would be appropriate.

AR: This statement has been removed.

31) P14 l21 – please rephrase "big"

AR: Done.

32) P14 l24 – please modify discussion of tuning. It is confusing to state that 5.5 and 5.4 were tuned for radiation but not for SST, so it is not surprising that they are worse, but they aren't that much worse. In the text it is suggested that the degradation of SST

is due to cloud forcing. Please modify this entire discussion. And please discuss why an untuned model is ready for reporting in the literature.

AR: This has been modified and revised to reflect the nature of the revised paper.